# Inference of neuronal functional circuitry with spike-triggered non-negative matrix factorization

Jian K. Liu[1,2], Helene M. Schreyer[1,2], Arno Onken [3], Fernando Rozenblit[1,2], Mohammad H. Khani[1,2,4], Vidhyasankar Krishnamoorthy[1,2], Stefano Panzeri [3] & Tim Gollisch [1,2]

Neurons in sensory systems often pool inputs over arrays of presynaptic cells, giving rise to functional subunits inside a neuron's receptive field. The organization of these subunits provides a signature of the neuron's presynaptic functional connectivity and determines how the neuron integrates sensory stimuli. Here we introduce the method of spike-triggered non-negative matrix factorization for detecting the layout of subunits within a neuron's receptive field. The method only requires the neuron's spiking responses under finely structured sensory stimulation and is therefore applicable to large populations of simultaneously recorded neurons. Applied to recordings from ganglion cells in the salamander retina, the method retrieves the receptive fields of presynaptic bipolar cells, as verified by simultaneous bipolar and ganglion cell recordings. The identified subunit layouts allow improved predictions of ganglion cell responses to natural stimuli and reveal shared bipolar cell input into distinct types of ganglion cells.

[1] Department of Ophthalmology, University Medical Center Göttingen, Waldweg 33, 37073 Göttingen, Germany. [2] Bernstein Center for Computational Neuroscience Göttingen, 37077 Göttingen, Germany. [3] Center for Neuroscience and Cognitive Systems, Istituto Italiano di Tecnologia, Corso Bettini 31, 38068 Rovereto, Italy. [4] International Max Planck Research School for Neuroscience, 37077 Göttingen, Germany. Correspondence and requests for materials should be addressed to T.G. (email: tim.gollisch@med.uni-goettingen.de)

Sensory systems often display strong signal convergence, with individual neurons pooling information over arrays of presynaptic connections. The characteristics of this signal pooling determine how the neuron responds to sensory stimulation and what type of computational role the neuron plays in information processing. A computational framework for analyzing the relation between functional connectivity and stimulus encoding is given by models that structure a neuron's receptive field into subunits, corresponding to the functionally relevant input channels. Such subunit models form the basis of our current understanding of, for example, retinal ganglion cell sensitivity to high spatial frequencies[1, 2], ganglion cell selectivity for specific types of motion signals[3–6], the emergence of orientation selectivity and phase invariance in primary visual cortex[7–13], and the processing of visual motion information along the cortical dorsal stream[14–16]. In the retina, ganglion cell subunits arise from nonlinear integration of bipolar cell signals[17–22]. Retinal subunit models have recently received increasing attention because they form the scaffold for specific computations performed by the retinal circuit[23, 24] and because of their apparent importance for understanding the encoding of natural stimuli[21, 25, 26].

However, connecting subunit models to concrete neuronal circuitry is complicated by the lack of methods that allow identification of the subunits from neuronal recordings. While receptive fields can be conveniently identified with white-noise stimulation and computation of the spike-triggered average[27], assessing the substructure within receptive fields has turned out to be a much harder problem. Efforts have therefore focused on fitting specifically constrained subunit models to data[10, 28–33] or by otherwise enforcing localized subunits in the receptive field[13, 34]. Furthermore, testing whether extracted subunits correspond to actual elements of the presynaptic circuitry provides an additional challenge, though progress can be made by comparing subunit characteristics with anatomical information[29]. Thus, methods that detect subunits of receptive fields with minimal prior assumptions about their number, size, or shape and with a demonstrated relation to functional connections in a neuronal circuit are highly desirable.

To this end, we here introduce a new method that we term spike-triggered non-negative matrix factorization (STNMF). The method identifies subunits in a way analogous to the identification of receptive fields through the spike-triggered average, that is, without the need to construct explicit models of the stimulus-response relation or to a priori specify the size, shape, number, or nonlinearity of the subunits. Furthermore, application of the method to recordings of retinal ganglion cells retrieves actual

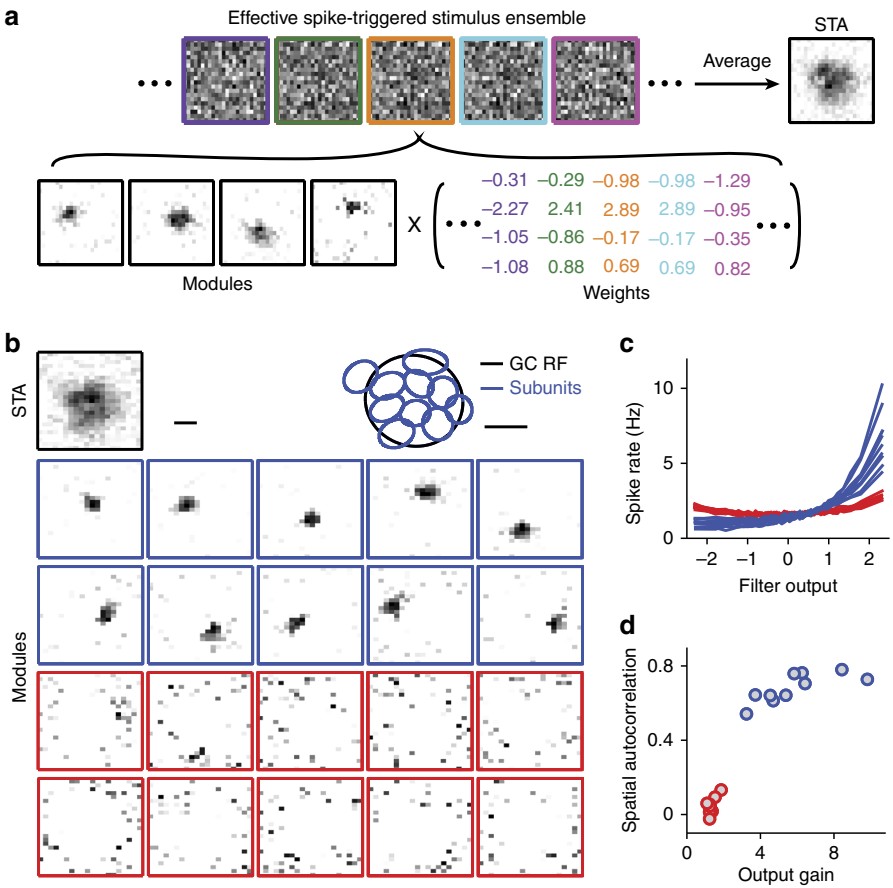

**Fig. 1** Identification of subunits with STNMF. **a** Illustration of STNMF analysis. Samples of a ganglion cell's effective spike-triggered stimulus ensemble (*top*), whose average corresponds to the cell's spike-triggered average (STA). For easier visual comparison with the subunits, STAs are here and in subsequent plots displayed with negative pixel values set to zero and with zero corresponding to *white* in the *greyscale* image. STNMF decomposes this ensemble into a set of modules and a weight matrix (*bottom*). The example here shows four modules that were identified for a sample ganglion cell. **b** Modules obtained for another sample ganglion cell by applying STNMF with 20 modules. Some modules have a strongly localized structure (*blue frames*), others are more noise-like (*red frames*). The *top row* shows the cell's receptive field, given by the spatial component of the STA, as well as the fitted receptive field outline (GC RF, *black ellipse*), together with outlines of the localized subunits (*blue ellipses*). Scale bars, 100 μm. **c** Nonlinearities for the 20 modules, with colors corresponding to the frame colors in **b**. **d** Quantification of spatial autocorrelation, obtained as Moran's *I*, vs. output gain, obtained as the spike-rate range of the nonlinearities shown in **c**, for the 20 modules. Colors as in **b** and **c**

receptive fields of presynaptic bipolar cells, thus providing a novel perspective on the functional connectivity and signal transmission between these successive neuronal layers.

## Results

**STNMF detects layouts of localized receptive field subunits.** We developed STNMF as a method for extracting the receptive field substructure that results from nonlinear pooling of functionally relevant inputs. To illustrate and explore the method, we analyzed responses of ganglion cells that we recorded from isolated salamander retinas with extracellular microelectrode arrays. The method only requires recorded ganglion cell spike times under spatiotemporal white-noise stimulation with fine spatial resolution. The core aspect is then to apply non-negative matrix factorization (NMF) to the collection of those stimulus patterns in the white-noise sequence that elicited spikes. NMF is a computational technique that is typically used to seek a decomposition of high-dimensional data into a relatively small set of modules and corresponding weights so that the individual samples in the data set are approximated by weighted combinations of the modules. When the data set is represented as a matrix of sample number vs. elements per sample, the decomposition amounts to a factorization of this matrix into one matrix that holds the set of modules and another matrix that holds the weights. Importantly, in NMF, this factorization is obtained under the constraint that the elements of all or some of these matrices are non-negative, which is known to facilitate the detection of sparse, parts-like modules[35]. This feature makes NMF attractive for trying to capture subunits, which can be viewed as the constituent parts of a receptive field. Intuitively, the modules derived by STNMF should capture the subunit decomposition of the receptive field because the spike-eliciting stimuli will have essential statistical structure imprinted on them by the subunits, such as correlations between pixel values, and the NMF method will make use of this structure to efficiently reconstruct these stimuli.

Concretely, we here proceeded as follows. For each analyzed ganglion cell, we extracted those 670-ms segments from the white-noise stimulus that preceded a spike. The collection of these spike-eliciting stimulus segments forms the spike-triggered stimulus ensemble, akin to common reverse-correlation analyses. For example, the average of all collected stimulus segments is the spike-triggered average (STA; Fig. 1a), a commonly used estimate of a cell's spatiotemporal receptive field[27]. To focus our analysis on the spatial structure of receptive fields, we collapsed each of the spike-triggered stimulus segments along the temporal dimension. We did so by computing an effective spike-triggered stimulus as the weighted average over time, where the weights were taken from the temporal profile of the cell's STA (Supplementary Fig. 1). Thus, the effective spike-triggered stimulus is a spatial pattern that measures how well the contrast sequence at each pixel matched the preferred temporal stimulus profile of the cell before a spike occurred, which essentially is a measure of how strongly each pixel was stimulated within this spike-triggered stimulus.

We then decomposed the ensemble of effective spike-triggered stimuli into a set of modules and a set of weights by applying semi-non-negative matrix factorization[36] (semi-NMF; Fig. 1a), a variant of NMF that is used here to implement the non-negativity constraint only for the elements of the modules. The weights, on the other hand, as well as the stimulus patterns were not constrained. Stimuli, for example, were represented by their contrast values, that is, by their relative deviations from mean light intensity, which could be positive or negative. Semi-NMF then seeks such weights and non-negative modules that minimize the difference, in a least-squares sense, between the spike-triggered stimuli and the reconstruction.

There is no analytical solution for the optimal modules and weights under the non-negativity constraint. Instead, we here used a numerical, iterative procedure (see Methods). Starting with a set of modules that are initialized with random numbers, we alternate between finding the best corresponding weights while keeping the modules fixed (which amounts to solving a standard least-squares problem) and optimizing the modules while keeping the weights fixed (which requires a complex, yet numerically tractable non-negative least-squares computation). In addition, to mitigate the risk of getting stuck in local minima during this iterative procedure, we periodically probe the effect of perturbing the current set of modules and weights by randomly deleting, duplicating, or splitting modules or adding noise (see Methods). This procedure eventually yields a stationary solution where no further improvements of modules and weights are found. The obtained final modules then indeed show structures that conform to the expectations for a layout of subunits. They are localized patterns of similar size that cover the ganglion cell's receptive field in an orderly manner (Fig. 1a, b).

In the application of non-negative matrix factorization, the number of considered modules must be specified before running the optimization procedure. This might suggest that this number needs to be individually optimized for each analyzed ganglion cell. However, we found that this is not the case. Instead, we can simply use a fixed number of modules, ideally somewhat larger than the expected number of subunits. As illustrated by a sample cell that was analyzed with 20 modules (Fig. 1b; see also Supplementary Fig. 2a, b for additional examples), it turns out that the method then yields modules of two types. Clearly, some of the modules show the localized structure expected for subunits, whereas the others look like noise. Indeed, when considering each of the modules as a spatial stimulus filter and relating the filter activation to the observed spiking activity in a histogram manner (Fig. 1c, see Methods), we found that the localized modules typically exert a strong effect on the cell's activity. The noise-like patterns, on the other hand, are largely uncorrelated with the occurrence of spikes, showing that they capture the noise in the spike-triggered ensemble. To quantify the effect of a module on the spiking activity, we computed the difference between the maximum and the minimum value in these histograms and called this the output gain of the subunit. Furthermore, we quantified how localized each of the obtained modules was by computing a measure of the spatial autocorrelation, called Moran's $I$ (see Methods). Thus, out of the set of obtained modules, the relevant subunits could be extracted in an automated manner according to their output gain and their level of localization (Fig. 1d). All further analyses of this work proceeded in this automated manner; STNMF was applied with 20 modules, and subunits were identified as those modules whose spatial autocorrelation or output gain crossed certain thresholds (see Methods).

As an aside, let us note that the computed output gain (Fig. 1d) provides a weight of how relevant a given subunit is for activating the ganglion cell. This can be interpreted as a measure of connection strength of the subunit to the ganglion cell. The relative weighting of the different subunits should be reflected in how strongly they contribute to the receptive field. Indeed, we found that the receptive field can be well fitted by the set of obtained subunits and that the weights of the subunits in this fit closely resembled their output gain values (Supplementary Fig. 2c, d). The same structure of relative importance of the subunits is also reflected in the average weights of the corresponding modules in the non-negative matrix factorization (Fig. 1a). The fact that these different measures of subunit contributions agree corroborates their interpretation as a measure of connection strength to the ganglion cell. This can be used, for example, to analyze the

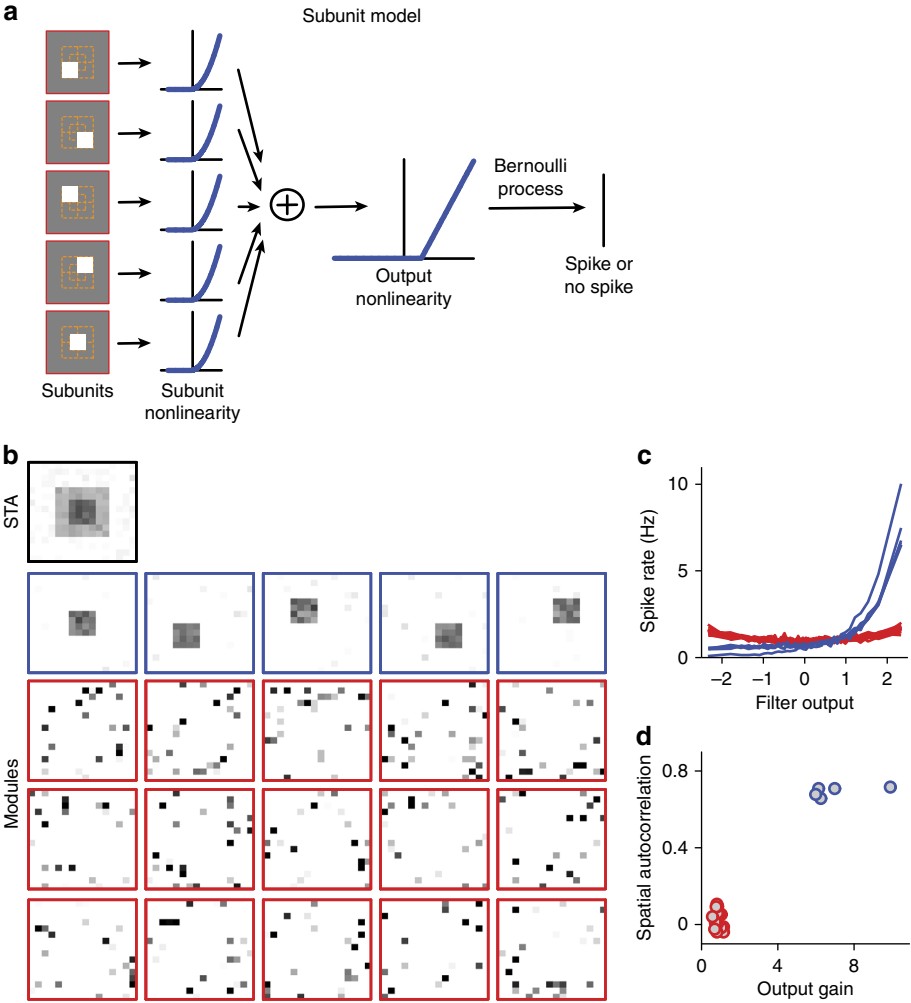

**Fig. 2** Identification of overlapping subunits with STNMF for a model neuron. **a** Model structure. Stimuli are here spatial patterns of 16 × 16 pixels of Gaussian white noise. The model contains five subunits, each covering a 4 × 4 region of the input space. The first four subunits tile the central 8 × 8 region of the input space; the last subunit is located in the center of the input space, overlapping partly with each of the other four subunits. The filter output of each subunit is transformed by a threshold-quadratic subunit nonlinearity before summation and application of a threshold-linear output nonlinearity with a positive threshold at unity to make spiking sparse, like in the observed recordings. This yielded a spiking probability that was used to determine actual spikes according to a Bernoulli process. **b–d**, STNMF analysis of simulated data, using 3500 spikes. STA as well as identified modules (**b**), nonlinearities corresponding to the modules (**c**), and analysis of spatial autocorrelation vs. output gain (**d**) are shown like in Fig. 1b–d. The analysis faithfully recovered the original subunits despite their spatial overlap

dependence of connection strength on the location of subunits within the receptive field, showing that the average subunit connection strength declined with distance from the receptive field center (Supplementary Fig. 2e). The weights of subunits will furthermore be important, as shown below, for combining the subunit signals into predictive models of ganglion cell activity.

A standard extension of the STA analysis for the case where multiple stimulus features are relevant is the method of spike-triggered covariance (STC) analysis[37, 38], which applies an eigenvalue analysis of the covariance matrix of the spike-triggered stimulus ensemble to detect relevant stimulus features. For comparison with the STNMF method, we therefore applied STC analysis to the ensemble of effective spike-triggered stimuli (Supplementary Fig. 3). The results indicate that (1) the STC analysis is more strongly impaired by the high dimensionality of the investigated stimulus space, allowing no clear separation between eigenvalues for relevant and non-relevant stimulus features, and that (2) features extracted by the STC analysis are not localized subunits, but rather correspond to a Fourier-like decomposition of the receptive field. This underscores the

importance of the non-negativity constraint in the STNMF for the extraction of localized subunits.

**STNMF robustly recovers true subunits of model neurons**. To check whether the STNMF analysis can indeed recover a known subunit layout, we tested the method on a simulated ganglion cell. The model cell contained five subunits that each filtered the incoming spatial stimulus patterns and then applied a threshold-quadratic rectification, akin to spatial nonlinearities observed for retinal ganglion cells[17]. The subunit signals were then summed by the ganglion cell (Fig. 2a). We here chose a subunit layout that contains considerable spatial overlap between the subunits, a scenario that is particularly challenging for subunit identification. Despite this overlap, we found that our approach faithfully recovered the true subunits of the model (Fig. 2b).

A subunit model requires that the subunit signals be combined in nonlinear manner; otherwise, the subunits would merge into a single filter and well-defined subunits would not exist. Yet, the exact shape of the subunit nonlinearity is not critical for subunit extraction with the STNMF method; for example, the subunits of

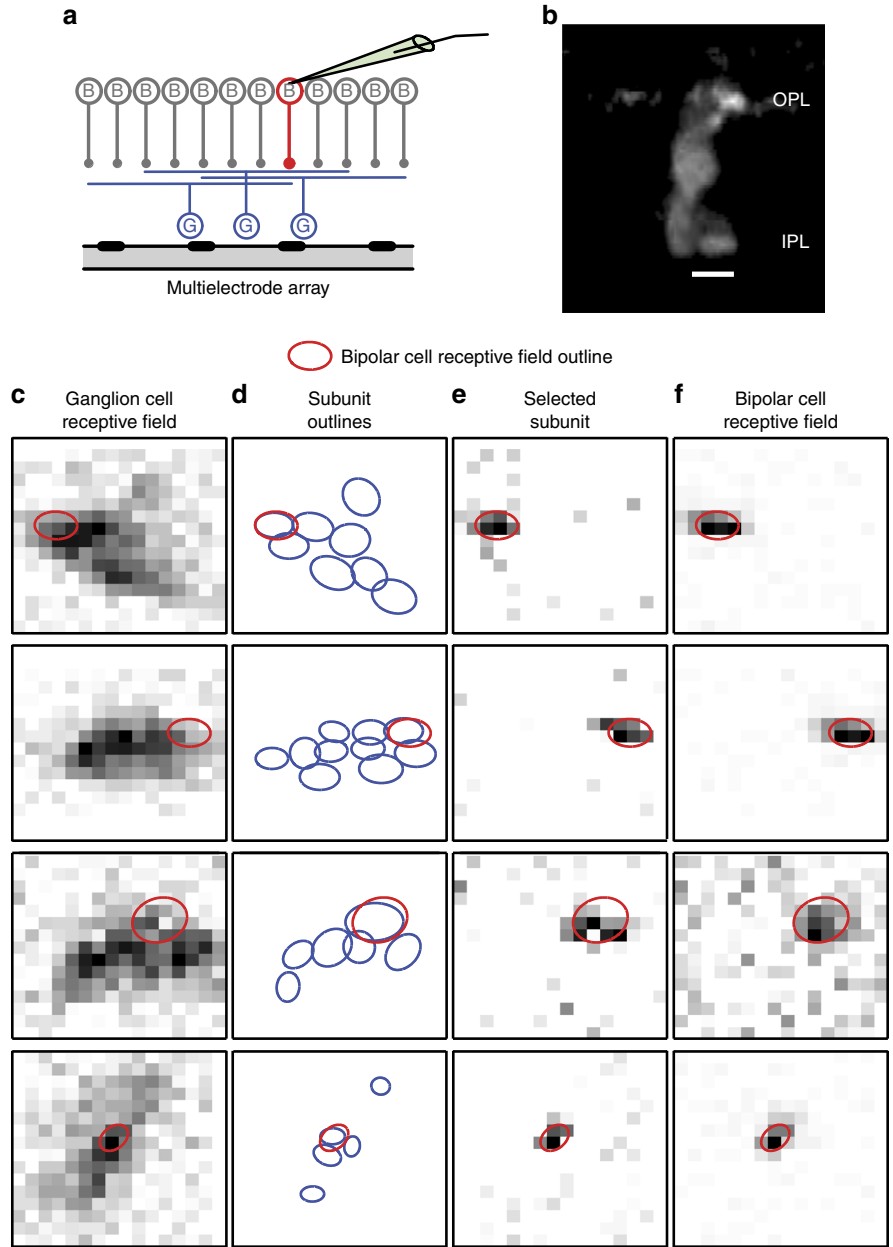

**Fig. 3** Comparison of ganglion cell subunits and simultaneously recorded bipolar cell receptive fields. **a** Illustration of the experiment (*left*), where one bipolar cell (B, *red*) is recorded with a sharp microelectrode, while multiple ganglion cells (G, *blue*) are simultaneously recorded with a multielectrode array. **b** Sample staining of a bipolar cell that was imaged after the combined intracellular and multielectrode-array recording, displaying neurites both in the outer plexiform layer (OPL) and inner plexiform layer (IPL). Scale bar, 10 μm. **c** Pixel-by-pixel representation of ganglion cell receptive fields, obtained as the spatial component of the spike-triggered average under spatiotemporal white noise. The *red ellipse* here and in subsequent panels shows the receptive field outline of the simultaneously recorded bipolar cell. **d** Outlines of identified subunits (*blue ellipses*). **e** Pixel-by-pixel representation of the best matching identified subunit. **f** Pixel-by-pixel representation of the bipolar cell receptive field, obtained as the spatial component from the corresponding reverse correlation analysis. Scale bar, 100 μm

the model were equally well identified when we replaced the threshold-quadratic rectification by a symmetric quadratic subunit nonlinearity (Supplementary Fig. 4). It is not surprising that the symmetric nonlinearity does not impede subunit identification because the subunits can contribute to the decomposition of the effective spike-triggered stimulus ensemble with either positive or negative weight (Fig. 1a). Note, though, that the average NMF weight of a subunit will now average out to near zero and therefore no longer provide a measure of connection strength of the subunit.

Because our described procedure begins with applying a single temporal filter to all spike-eliciting stimuli, one may wonder whether the approach fails when temporal filtering differs between the subunits. However, this is not the case as shown by analysis of a spatiotemporal model where each subunit was associated with a different temporal filter. This did not prevent the method from recovering the true subunits (Supplementary Fig. 5). The intuition behind this is that as long as there is some overlap between the individual temporal subunit filters and the temporal component of the cell's STA, the effective spike-

triggered stimuli will still reflect the spatial structure that is imposed by the subunits.

How robust is the subunit identification with respect to noise and to the amount of data available? To explore this, we first used the model of Fig. 2 and varied either the level of noise or the number of included spikes (Supplementary Fig. 6a–c). The former was done by replacing a fraction of the simulated spikes, chosen at random, with spikes at random times and the latter by successively including more simulated spikes in the analysis. We measured performance by computing the correlation between the true model subunits and the corresponding reconstructed modules. In addition, since the true subunits are characterized by a high spatial autocorrelation, we also measured performance

by the average spatial correlation of the five most localized modules. This measure has the advantage that it can also be applied when the true subunits are not known. As expected, reconstruction of the subunits deteriorated with increasing noise, yet small levels of noise only had a relatively mild effect, and only once about half the spikes had been replaced did performance decline steeply. Similarly, the subunit reconstruction was fairly robust with respect to spike number; here, performance leveled off once about 1500 spikes were included. A similar qualitative dependence of subunit reconstruction on noise and spike number was observed in sample ganglion cells (Supplementary Fig. 6d–i), indicating again a certain robustness to noise and good reconstruction performance once several thousand spikes were

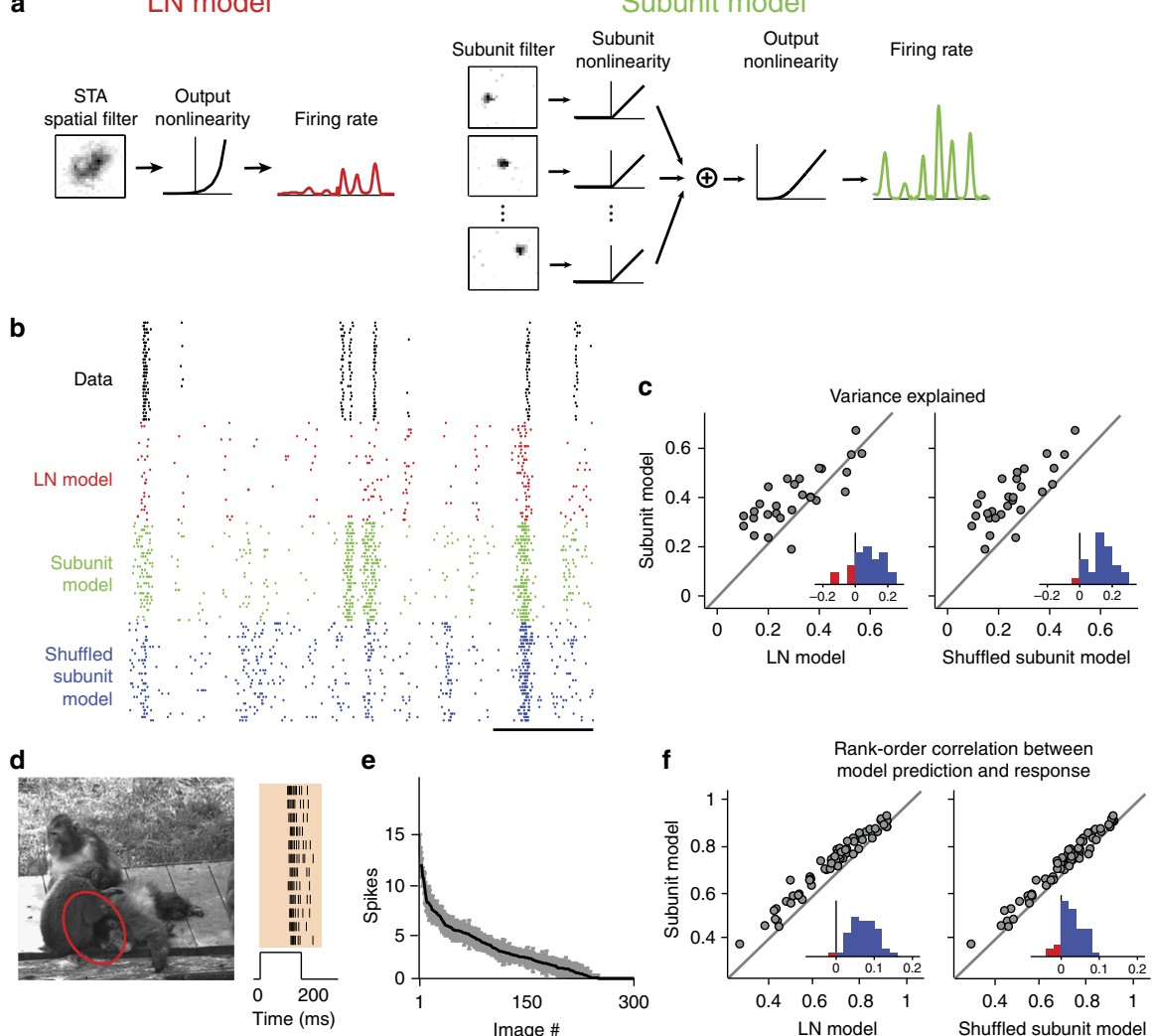

**Fig. 4** Using subunits to predict ganglion cell responses. **a** Illustration of model structures for the linear-nonlinear (LN) model (*left*) and the subunit model (*right*). The subunit model takes each identified subunit as a spatial filter and then sums their outputs after rectification. Both models pass the obtained filter signal through a final output nonlinearity to predict the cell's firing rate. **b** Raster plots of measured spikes and simulations of the fitted models for a sample cell under frozen spatiotemporal white noise. Spikes were simulated by a Poisson process that takes the models' firing rates over 33-ms bins as input. Scale bar, 1 s. **c** Comparison of variance explained over the frozen-noise section for the different models. Each data point corresponds to a different ganglion cell ($N = 28$ cells from one retina). *Insets* show histograms of differences in variance explained. The variance explained is significantly higher for the subunit model compared to the LN model ($p = 9 \times 10^{-4}$; Wilcoxon signed-rank test) as well as compared to the shuffled subunit model ($p = 1 \times 10^{-5}$). **d** Sample image used to measure responses to briefly flashed natural images and corresponding measured spike responses from a sample ganglion cell for several stimulus repeats. The red ellipse shows the 3-sigma outline of the ganglion cell receptive field. The *shaded region* in the raster plot shows the window over which spikes were counted. **e** Average spike counts (*black line*) and standard deviations (*grey region*) of the sample cell for each of the 300 images. **f** Comparison of rank-order correlations between predicted and measured responses for the different models. Each data point corresponds to a different ganglion cell ($N = 46$ cells from six retinas). *Insets* show histograms of differences in rank-order correlation. Differences between the subunit model and the other two models are significant ($p < 10^{-6}$ in both cases; Wilcoxon signed-rank test)

included in the analysis. Moreover, inspecting the modules obtained under different levels of noise and spike numbers suggests that, when the method starts failing, it does so by first losing individual subunits while faithfully retaining other subunits.

**Detected subunits match bipolar cell receptive fields**. Given the reported role of bipolar cell inputs for nonlinear spatial integration by ganglion cells[18–21], we hypothesized that the subunits identified by the STNMF method from ganglion cell recordings corresponded to individual presynaptic bipolar cells. To check this hypothesis, we combined the multielectrode array recordings with simultaneous recordings from individual bipolar cells through sharp microelectrodes. Indeed, we often found that the bipolar cell receptive field closely matched one of the subunits from one or more ganglion cells (Fig. 3; see also Supplementary Fig. 7 for more examples).

We quantitatively evaluated the match of bipolar cell receptive fields and subunits by computing their overlap, defined as the relative shared area (see Methods). From a total of 17 recorded bipolar cells, we found 9 bipolar cells whose receptive field displayed a close match with at least one ganglion cell subunit, as shown by overlap values >0.5 and reaching up to 0.86. For other recorded bipolar cells, we found no matching subunits (Supplementary Fig. 7), consistent with the expectation that not all ganglion cells will be connected to a given nearby bipolar cell because of the multitude of bipolar and ganglion cell types.

To statistically evaluate these overlap values, we compared them to maximal overlap values from surrogate data sets, which we obtained by randomly rotating the receptive fields and randomly permuting their center positions for all ganglion cells whose receptive fields overlapped with a given bipolar cell (see Methods). This procedure showed that the experimentally observed maximal overlap values were significantly larger than the maximal overlap values that should be encountered by chance ($p = 0.0068$, Fisher's combined probability test). Thus, the good match of bipolar cell receptive fields and ganglion cell subunits suggests that the subunits identified with the STNMF method indeed correspond to the receptive fields of individual presynaptic bipolar cells.

**Subunit layouts help predict ganglion cell activity**. Are the STNMF-derived subunits relevant for understanding how the ganglion cells respond to visual stimuli? We studied this question by assessing the importance of the subunits for predicting the ganglion cell response to novel stimuli. We first considered "frozen-noise" sections from the spatiotemporal white-noise stimulation that were repeatedly inserted into the stimulus sequence. The frozen-noise sections were not used for estimating the subunits and therefore provided a held-out test set. Temporal filtering was again taken care of by convolving the stimulus sequence of each pixel with the temporal component of the cell's STA, analogous to the computation of the effective spike-triggered stimuli (Supplementary Fig. 1). We then compared a standard linear-nonlinear (LN) model, where the spatial component of the STA was used as a stimulus filter, to a subunit model that contained multiple filters, corresponding to the identified subunits (Fig. 4a). In the subunit model, the filter outputs were rectified and then summed in a weighted manner. The weights were determined by the linear combination of subunits that best matched the receptive field. This means that the LN model and the subunit model gave approximately equal total weight to each stimulus pixel. For both models, a subsequent output nonlinearity, obtained from the non-repeated sections of the white-noise stimulus, transformed the resulting signals into a firing rate.

Compared to the LN model, we found that this simple approach of incorporating the subunits could already substantially improve the prediction of firing rates for the held-out spatiotemporal white-noise segments, in particular when the LN model itself performed poorly (Fig. 4b, c). Intuitively, the superior performance of the subunit model results from the fact that pixel values will often cancel out toward zero over the entire receptive field, thus not allowing the LN model to predict spikes. Individual subunits, on the other hand, may still be strongly activated, leading to spiking responses both in the subunit model and in the experimental data.

One might hypothesize that the superior performance of the subunit model results primarily from inserting an intermediate rectification and not from including the layout of localized subunits. To test this hypothesis, we also set up a shuffled subunit model. The shuffled subunit model was obtained by taking the subunits of the subunit model and randomly permuting the pixel values at each spatial location across the subunits. Otherwise, the shuffled subunit model was treated and evaluated in the same way as the actual subunit model. Comparing performance of these two models showed that shuffling the subunits leads to substantially worse response predictions (Fig. 4c, *right*), indicating that, beyond the rectification before spatial summation, the actual subunit layout is important for response predictions under white-noise stimulation.

To evaluate whether the subunit layout is also important for predicting responses to natural stimuli, we measured ganglion cell responses to briefly flashed natural photographs (Fig. 4d, e) and compared these to model predictions. We only considered the average spike count elicited by each image and could therefore omit temporal stimulus integration here. We furthermore dispensed with the output nonlinearities for this analysis and only aimed at predicting the rank order of the 300 applied images. This made the model comparison independent of how well the final output nonlinearity captures the neuronal sensitivity in this specific stimulus context. Thus, the only model parameters here were the receptive field for the LN model and the set of subunits for the subunit model, which were all obtained from the spatiotemporal white-noise experiments. Similar to the results under white-noise stimulation, we found that the subunit model clearly and systematically outperformed both the LN model as well as the shuffled subunit model (Fig. 4f), demonstrating the importance of the subunit layout for capturing the encoding of natural stimuli. Likewise, we found that the subunit model was also superior in capturing more subtle differences between images; it predicted response differences for slightly shifted presentations of the same natural images much better than the considered alternatives (Supplementary Fig. 8).

While these analyses showed that knowledge about the subunits can improve response predictions over the simple LN model, there is apparently still room for improvement, in particular, through optimizing the subunit nonlinearities and including feedback dynamics both on the level of subunits and on the global level after subunit integration. We here did not pursue these options further because our primary goal in this work was to investigate the potential of the STNMF method to extract subunits that are interpretable in terms of actual circuit elements. Yet, to check how the effect of including subunits relates to current state-of-the-art approaches in analyzing nonlinear stimulus integration, we compared our results to the Nonlinear Input Model (NIM), which extends the Generalized Linear Model framework to include subunit-like nonlinear stimulus integration[28]. In order to allow for a direct comparison with our subunit model and to cope with the high dimensionality of our stimulus

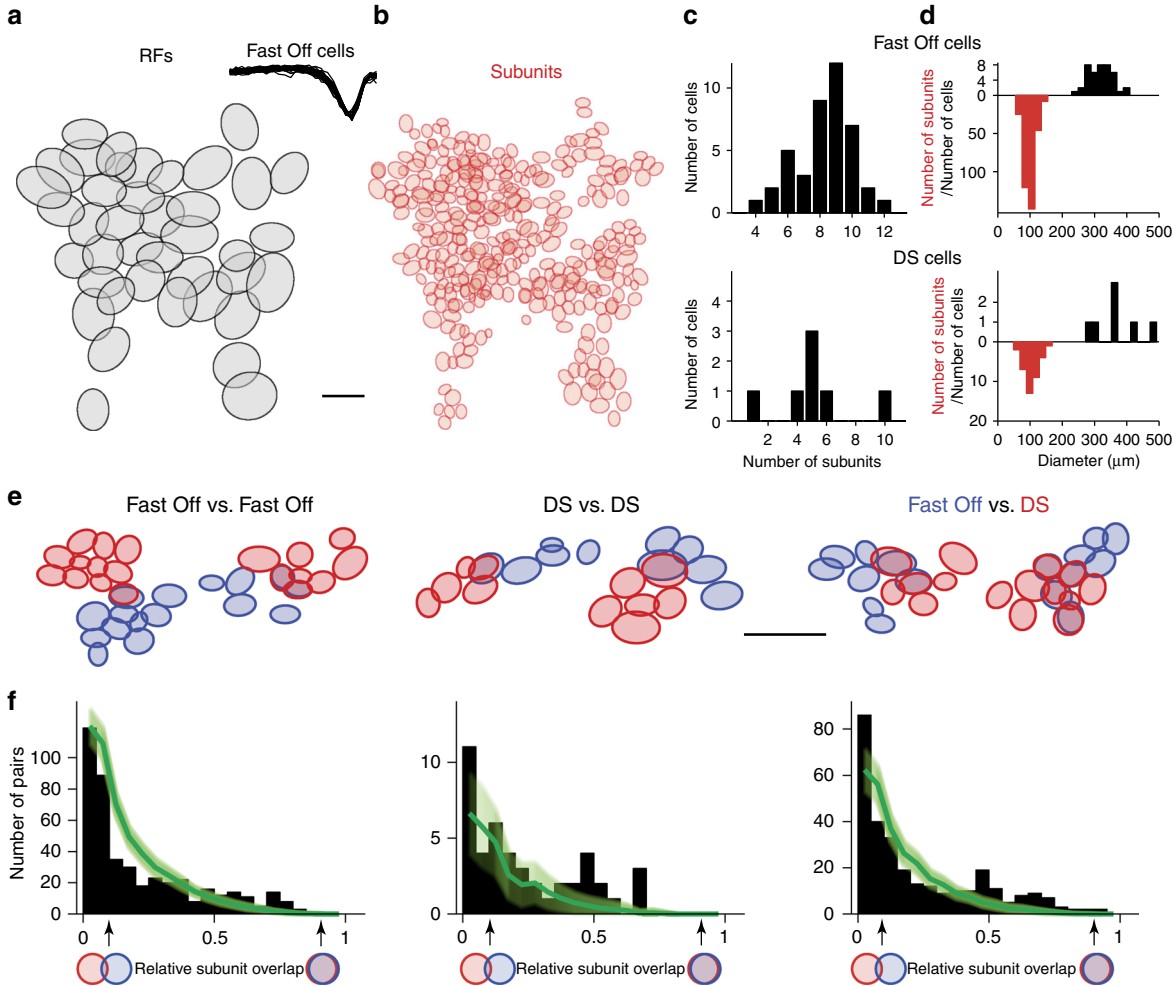

**Fig. 5** Analyzing subunit overlap for ganglion cell populations. **a** Layout of receptive fields of fast Off ganglion cells from a single recording. Scale bar, 300 μm. *Inset* shows the collection of temporal filters (670 ms long) for the fast Off cells. **b** Layout of reconstructed subunits for the collection of cells shown in **a**. **c** Histograms of the numbers of subunits per cell for fast Off and direction-selective (DS) ganglion cells from the experiment shown in **a**. **d** Histograms of the diameters of receptive fields (*black*) and subunits (*red*). **e** Examples and distributions of overlapping subunits among fast Off cells (*left*), DS cells (*center*), and pairs of fast Off and DS cells (*right*). Scale bar, 300 μm. **f** Distributions of relative overlap for pairs of subunits from different ganglion cell classes (*black bars*), collected from five retinas (110 fast Off cells, 40 DS cells). Subunit pairs with zero overlap were excluded. For comparison, distributions of chance overlaps, obtained by shuffling the receptive field center positions within each ganglion cell class, are shown in green (*shaded region* corresponding to one standard deviation, obtained from repeated shuffling). All actual overlap distributions show an excess of well matching subunits (relative overlap larger than about 0.5) as compared to chance, indicating that ganglion cells within as well as across the two ganglion cell classes share input from the same bipolar cells

space, we here used a reduced version of the NIM, with time integrated out like for the STNMF approach and no dependence of spiking probability on previous spikes. For this reduced NIM, we found that the obtained filters, unlike the subunits of the STNMF method, were not localized in space and that the response predictions for held-out white noise segments did not improve over the STNMF-derived subunit model (Supplementary Fig. 9). Clearly, we have not explored the full potential of the NIM here, but the comparison at least indicates that subunits obtained by STNMF provide a competitive spatial layout for developing generative models of ganglion cell spiking.

**Identification of shared input to different ganglion cells**. The STNMF analysis does not require adjusting stimuli for the analysis of individual cells. We can therefore obtain subunits from many ganglion cells recorded simultaneously under spatiotemporal white-noise stimulation and compare the layout of identified subunits for cells with overlapping receptive fields. This

provides for the possibility to check whether ganglion cells of different identified functional types have matching subunit layouts, which would indicate that they share presynaptic inputs from the same bipolar cells. In our recordings from the salamander retina, two classes of ganglion cells could be readily identified (see Methods). Fast Off cells were characterized by their relatively small receptive fields, fast temporal filters and stereotypic spike autocorrelation, and their receptive fields were found to tile the retina (Fig. 5a). Direction-selective (DS) ganglion cells were detected by their directional tuning for drifting gratings (see Methods), though the class of DS cells may contain DS cells of two different types[39].

The STNMF method allowed subunit identification in a high-throughput manner for each cell in these large populations, providing a dense map of bipolar cell inputs to fast Off cells (Fig. 5b), with about 8–10 identified subunits per cell (Fig. 5c). For both fast Off and DS cells, the sizes of identified subunits were densely clustered around 100 μm diameter (Fig. 5d). This matches previous measurements of bipolar cell receptive field

sizes in the salamander retina[40], which had yielded diameters in the range of 50–150 μm, and is also comparable to typical salamander bipolar cell dendritic field sizes[41].

The dense sampling of analyzed ganglion cells allowed us to investigate the relative positioning of subunits for pairs of ganglion cells within and across the two cell classes. For pairs of both fast Off and DS cells as well as for mixed pairs, we found that the cells can display subsets of closely matching subunits (Fig. 5e). In order to analyze whether these matches indicate a systematic structure of the subunit inputs to multiple ganglion cells or whether they could simply be chance occurrences, we evaluated the overlap between all pairs of subunits from different ganglion cells in the same recordings. To do so, we computed the relative overlap of a subunit pair as the ratio between the shared area and the total area covered by the 1.5-sigma contours of the subunits. We then compiled these values into histograms (Fig. 5f; leaving out subunit pairs with no overlap at all) and compared them to overlap distributions from randomized subunit layouts. The latter were obtained by repeatedly permuting the spatial positions of the ganglion cells at random, that is, for each ganglion cell, we spatially translated the subunit layout so that the receptive field center position of that cell was shifted to the receptive field center position of a randomly chosen ganglion cell of the same class in the same recording. This showed that the true ganglion cell layout yielded many more pairs of subunits with a large relative overlap than the randomly permuted ganglion cell layouts. Note that large subunit overlaps by chance were rare; the distributions obtained from the random permutations rapidly approached zero for increasing overlap values. Each of the actual overlap distributions, on the other hand, contained many pairs of subunits with overlap values >0.5, corresponding to several times the number of subunit pairs expected from chance (fast Off vs. fast Off: 69 actual pairs vs. 25 ± 5, mean ± SD, in the shuffle data; DS vs. DS: 6 vs. 1 ± 1; fast Off vs. DS: 49 vs. 11 ± 4). Thus, the STNMF-derived subunits capture the systematic sharing of bipolar cell inputs by ganglion cells of the same type as well as across the two analyzed classes of ganglion cells. Furthermore, it is noteworthy that the strongly overlapping subunits of different ganglion cells were recovered from independent sets of spikes, providing further evidence that the identified subunits indeed correspond to specific circuit elements, namely presynaptic bipolar cells.

## Discussion

We here demonstrated that STNMF can reveal the subunit structure of receptive fields (Figs. 1 and 2). An appealing aspect of the method is that the subunits naturally emerge from the analysis—much like the receptive field emerges from an STA analysis—without the need to guess a specific model structure that may be constrained in terms of subunit shape or subunit nonlinearity or that may require identical subunits arranged in an orderly layout. Moreover, we showed that the identified subunits match actual circuit elements, the receptive fields of presynaptic bipolar cells (Fig. 3). Thus STNMF provides a way to link structure and function in the neural circuit, for example, by using the inferred subunit layout as a basis for improved models of sensory encoding (Fig. 4). Finally, application of the method only requires recorded spike times under high-resolution white-noise stimulation and is therefore applicable to high-throughput analyses of populations of simultaneously recorded cells (Fig. 5), providing insight into the divergence and convergence of sensory information between subsequent processing layers.

Regarding bipolar-to-ganglion cell connectivity, previous studies, based on current injections into bipolar cells[42] as well as anatomy-based circuit reconstructions[43], had suggested

considerable diversity in the connectivity matrix between bipolar and ganglion cells. Here, on the other hand, the identified orderly sets of subunits within ganglion cell receptive fields suggest a rather high degree of specificity in the functional connectivity of individual ganglion cells. It seems likely that the identified subunits correspond to primary bipolar cell connections, which provide the major excitatory input under the applied stimulus, but that there may also be secondary connections, including polysynaptic connections[42], with other bipolar cells, which may have more modulatory effects or are invoked in other stimulus contexts. Furthermore, the functionally rather different types of fast Off and DS cells in the salamander retina appear to share inputs from the same bipolar cells. That such input sharing exists should be expected simply because of the larger number of ganglion cell types compared to bipolar cell types. Indeed, similar sharing was seen in anatomical investigations of mouse retina, with DS cells strongly connecting to the same bipolar cells as local edge detectors[43]. Analyses of subunit sharing as presented here complement these connectivity studies by identifying common input from a functional perspective.

How does STNMF work? Essentially, the method detects correlation patterns in the set of spike-eliciting stimuli. Stimulus pixels within a subunit can more easily cancel each other's effect on the activation of the ganglion cell than pixels from different subunits, and their contrast values will therefore be more strongly correlated for spike-eliciting stimuli to avoid this cancellation. These correlations are extracted and used by the NMF to find a basis for representing the spike-eliciting stimuli. Thus, the method, as introduced here, is restricted to stimuli with white-noise statistics because correlated stimuli would bias the NMF to represent these prior stimulus correlations. Also, this shows that the nonlinear integration of subunit signals is essential because this nonlinearity creates the differences in the pixel correlations. Thus, the method would fail for cells with purely linear stimulus integration, and stronger nonlinearities (typically stronger rectification) will facilitate the subunit detection. Note also that the non-negativity constraint on the modules prevents the method from identifying opposing contrast preference within the same subunit. Thus, the identified subunits may be thought of as receptive field centers of bipolar cells, but any bipolar cell receptive field surround would not be detected. A further limitation is the dependence on sufficient amounts of data. We here generally aimed for recording several thousands of spikes under the white-noise stimulation to have enough statistics in the spike-triggered ensemble. A typical effect of insufficient data is the failure to detect all subunits so that parts of the receptive field remain uncovered by subunits. Owing to the data demand, potential future extensions of the method that may aim at analyzing full spatiotemporal subunits need to find ways to handle the resulting higher stimulus dimensionality. This may be facilitated by combining NMF with a factorization into sets of purely spatial and purely temporal modules, an approach that has recently been developed for decomposing neuronal population activity[44].

We tested the STNMF method on ganglion cell recordings from salamander retina, a system that allowed us to compare the extracted subunits with simultaneously recorded bipolar cell receptive fields. Yet, the method should also work similarly for mammalian retinas. In fact, we found in a proof-of-principle manner that application of STNMF to ganglion cell recordings from mouse retina also yields orderly layouts of localized subunits (Supplementary Fig. 10).

Moreover, nonlinear pooling of presynaptic signals is ubiquitous in sensory systems, and identifying underlying subunit structures is a common goal not only for the retina[20, 21, 29, 34], but also for thalamus[28], primary visual cortex[10–13], higher visual

cortex areas[14–16], and the auditory system[28, 45]. The STNMF method should be readily applicable to these different systems, as it does not require specific assumptions about the nonlinear interactions between subunits and does not rely on concomitant acquisition of anatomical information[20, 46]. Note that the spatial stimulus dimension that was analyzed here could be replaced by other stimulus coordinates, along which neurons might be organized, such as spectral components of auditory stimuli.

Future applications of the STNMF method may lie in investigating neural coding as well as in analyzing neural circuits. For example, subunit models enhanced by further features, such as optimized subunit nonlinearities, inhibitory interactions, and/or adaptation or feedback dynamics both before and after spatial summation should provide a promising basis for the difficult task of modeling responses to general, natural stimuli[26]. Using STNMF as a designated tool to first obtain the subunit layout with subsequent investigations of additional model features will facilitate parameter estimation for such enhanced models. For example, a previous exploration of a retinal subunit model with temporal filtering and feedback[30], which applied a global optimization approach for all model parameters, had to restrain the parameter space by enforcing identical shapes of subunits and restricting the analysis to a single spatial dimension by applying a stripe-like stimulus layout. Prior identification of subunits by STNMF may considerably reduce the numbers of parameters in a subsequent model fit. Moreover, when subunit identification is performed in an online fashion during an experiment, probe stimuli could directly be targeted at individual subunits in order to examine local nonlinearities[20, 29] as well as local adaptation features[47, 48].

Similar analyses might also be applied when a retina is not stimulated via its photoreceptors, but through optogenetic constructs inserted into bipolar cells[49–51], which is currently considered as a promising approach for vision restoration therapy when photoreceptors have degenerated. Subunit identification in such therapeutic models should aid analyses of how the optogenetic stimulation is processed by the retina and how it might be optimized to activate ganglion cells. Finally, large-scale recordings from the retina may allow functional identification of a large set of different ganglion cell types[52], providing a basis for large-scale analyses of functional connectivity, complementing high-resolution measurements of receptive fields[53] as well as anatomy-based identification of connection patterns[43, 54].

## Methods

**Electrophysiology.** Recordings were made in isolated retinas obtained from axolotl salamanders (*Ambystoma mexicanum*; pigmented wild type) of either sex. All experimental procedures conformed with institutional guidelines of the University Medical Center Göttingen. Multielectrode array (MEA) recordings of ganglion cell spiking activity were obtained as described previously[55]. In brief, animals were killed after about 30 min dark adaptation, and both eyes were enucleated and hemisected to isolate the retina. The retina was placed onto a 252-channel MEA in a recording chamber and superfused with oxygenated Ringer's solution at room temperature (around 22 °C). A transparent dialysis membrane stretched across a plastic holder was placed on top of the retina to hold it in place during the recording. We also performed MEA recordings with retinas from adult wild-type mice (C57BL/6, 7–12 weeks old) of either sex as described previously[56]. Here, retinas were superfused with oxygenated Ames' medium, buffered with 22 mM NaHCO₃, heated to a constant temperature around 33–34 °C, and held in place by a coating of poly-D-lysine on the array. Spikes were extracted by custom-made software, based on a Gaussian mixture model and an expectation-maximization algorithm[57]. Only well-sorted units with clear refractory period were used for further analysis.

Combined recordings of ganglion cells and bipolar cells from salamander retina were obtained by placing the retina onto a 60-channel perforated MEA[58]. The perforated MEAs contain small holes in between the electrodes, so that slight suction applied underneath the array held the retina tightly in place while allowing unrestricted access to the tissue with sharp microelectrodes from the top. Sharp microelectrodes, tip-filled with 4% Neurobiotin, dissolved in 0.1 M Tris buffer, and backfilled with 3 M KCl solution with impedance in the range of 100–300 MΩ,

were inserted blindly into retinal cells. By monitoring through a 60× objective, the microelectrode was positioned at the photoreceptor surface of the retina roughly above a selected MEA recording site, which was chosen according to the observed ganglion cell spiking activity to maximize chances of recording pairs of bipolar and ganglion cells with overlapping receptive fields. The microelectrode was then inserted into the retina and advanced until a cell was impaled while monitoring the depth of the electrode tip in the tissue. After the recording, Neurobiotin was injected into the intracellularly recorded cell with current pulses (alternating blocks of several seconds of 1 Hz positive or negative pulses). The retina was fixated with 4% paraformaldehyde, processed first with Alexa Fluor 488 Streptavidin, and imaged under a confocal microscope to visualize the Neurobiotin-filled cell. Two-dimensional images were obtained by a maximum projection in the x–z plane. To reduce blur, we applied the blind-deconvolution algorithm in Matlab (deconvblind), using an estimate of the point-spread function obtained by imaging fluorescent beads.

When cell staining and imaging were successful, the cell morphology was used to confirm that the intracellularly recorded cell was indeed a bipolar cell. Occasionally encountered amacrine cells were discarded for the present analysis. In some experiments, morphological reconstruction failed because the recorded cell was lost before dye injection or because the retina could not be removed from the perforated MEA in an intact manner[58]. We then used additional criteria to identify bipolar cells. Bipolar cells were distinguished from photoreceptors by recording depth, by the receptive field size, and by the characteristic light responses of photoreceptors. Intracellularly recorded cells that were morphologically verified as photoreceptors showed stereotypical response traces to positive and negative contrast steps and receptive field sizes of 10–20 μm diameter. In some recordings, we also applied the AMPA/kainate antagonist CNQX at the end of the experiment to check that light responses were lost for putative Off-type bipolar cells, whereas photoreceptor light responses were insensitive to CNQX. To distinguish bipolar cells from amacrine cells, we applied depolarizing and hyperpolarizing current pulses into the recorded cell (50–500 pA, 500 ms duration, 2 s intervals) and checked that a robust activation of ganglion cells to positive current injection was observed[59].

**Visual stimulation.** Visual stimuli were controlled by custom-made software, written in C++ and using the OpenGL library, and displayed on a gamma-corrected, monochromatic white OLED monitor (800 × 600 pixels, 60 Hz refresh rate). The monitor was projected onto the photoreceptor layer either through a telecentric lens above the retina (252-electrode MEA recordings, pixel size 7.5 × 7.5 μm on the retina) or through the objective of an upright microscope with a beamsplitter inserted into the light path (combined bipolar cell and MEA recordings, pixel size 2.5 × 2.5 μm on the retina).

We optically stimulated the retina with spatiotemporal white noise, temporally updated at a rate of 30 Hz and spatially arranged in a checkerboard layout with stimulus pixels of 30 × 30 μm. For some bipolar cell recordings, the stimulus update rate was reduced to 10 or 15 Hz. The light intensity of each square was chosen independently from a binary distribution with 100% contrast and a mean light level of about 2.5 mW m⁻². In all analyses, stimuli were represented by their contrast values, so that the stimulus values for the bright and dark squares were +1 and −1, respectively. Typical recording durations with this stimulus were 60–180 min for standard MEA recordings and 60 min for combined bipolar/MEA recordings. For bipolar cell recordings, we also applied 10 min of stimulation with spatiotemporal white noise of 10 × 10 μm pixel size to evaluate receptive field size for small cells and distinguish bipolar cells from photoreceptors.

To stimulate the retina with natural images, we selected a set of 300 natural photographs from the McGill Calibrated Colour Image Database[60], displaying a wide range of natural scenes. Each image had a spatial resolution of 256 × 256 pixels, covering a total area of 1920 × 1920 μm on the retina. The provided RGB-color values for each image were converted into greyscale by a weighted average of the three color channels with R:G:B = 30:59:11. Mean and standard deviation of the pixel values were normalized for each image by appropriately shifting and scaling the values so that the mean pixel intensity was equal to that of the spatiotemporal white noise, and the standard deviation was 50% of the mean intensity. Pixel values that then deviated from the mean by more than 100% in either direction were clipped to ensure that the maximal pixel values were within the physically available range of the display. To minimize the artefacts induced by this clipping, we selected images that had only few clipped pixels (not more than 0.1% of the pixels). Images were presented individually for 200 ms each in a pseudo-random sequence, with an inter-stimulus-interval of 800 ms, during which uniform illumination at the mean intensity was presented. For data analysis, we counted the number of spikes for each ganglion cell over a 300-ms window following stimulus onset.

For experiments with shifted natural images, we further selected 10 images from the set of 300 images. Each image was presented with nine different center positions, arranged on a 3 × 3 square grid, with 90 μm between adjacent positions. Stimulation with the resulting 90 images and assessment of responses then occurred in the same way as for the set of 300 different natural images.

**Receptive field analysis.** From the response to the spatiotemporal white-noise stimulus, we computed the STA for each recorded ganglion cell[27], taking into account stimulus sequences of 670 ms before each spike. We then decomposed the

STA into a temporal filter and a spatial receptive field through singular-value decomposition[61], using the top-rank temporal and spatial components of the decomposition. Both temporal filter and receptive field were normalized to unit Euclidean norm so that the sum of squares of the filter elements equaled unity for both components. Receptive field sizes were estimated by fitting a two-dimensional Gaussian function to the spatial receptive field and determining the effective diameter $d = \sqrt{a \cdot b}$, where $a$ and $b$ are the major and minor axis of the 1.5-sigma contour of the fitted Gaussian.

For bipolar cells, the same approach was used, except that the STA was replaced by the reverse correlation of the stimulus with the measured membrane potential. Specifically, we temporally binned the recorded membrane potential at the same rate as the stimulus update rate by computing the average potential over each bin. We then subtracted the mean from this sequence of membrane potentials and used the resulting values as weights for the preceding stimulus sequences. The weighted sum of the stimulus sequences then provided the spatiotemporal receptive field of the bipolar cell.

**Cell-type classification**. We detected DS ganglion cells by analyzing the responses to drifting square-wave gratings (100% contrast, 600 μm spatial period, 450 μm s$^{-1}$ velocity) moving in eight different directions for 6.67 s each, with the entire sequence repeated five times. From the average firing rate $r_\theta$ for each direction $\theta$ (leaving out the first 1.33 s after stimulus onset), we computed a direction-selectivity index (DSI) as the normalized vector sum, $\text{DSI} = \left| \sum_\theta r_\theta \, e^{i\theta} \right| / \sum_\theta r_\theta$. Note that the definition of the DSI via the vector sum differs from the often applied measure that uses the response difference between preferred and null direction. We then determined DS cells as cells with a significantly large DSI and at least DSI > 0.25. Significance here was established by repeatedly shuffling the firing rates over all angles and trials 10,000 times for a given cell to obtain the distribution of DSI values under the null hypothesis that the firing rates are independent of the motion direction. A cell for which the DSI was larger than 95% of the DSI values obtained from the corresponding shuffled data was considered significantly direction selective.

Note that two types of DS ganglion cells have recently been found in the salamander retina[39], "standard DS cells" and "OMS-DS cells". The latter are characterized by their small receptive fields and by also being object-motion-sensitive (OMS), that is, their responses are (partly) suppressed under global motion[4]. We here did not distinguish between these two types of DS cells, yet we generally find considerably more standard DS cells than OMS-DS cells in our recordings[39], and the fairly large receptive field sizes of the DS cells in the present data (364 ± 73 μm, mean ± SD) also indicate that most of the analyzed DS cells are of the standard DS cell type.

Fast Off cells were identified by performing a cluster analysis of simultaneously recorded ganglion cells. To obtain the parameters for the cluster analysis, we performed principal component analysis on the set of all temporal filters as well as on all spike-train autocorrelation functions. The latter were obtained also under spatiotemporal white noise, with a maximal lag of 60 ms and binned at 2-ms resolution. Each cell was then described by four parameters, its receptive field size, the projections of its temporal filter on the first two principal components of all temporal filters, and the projection of its autocorrelation function on the first principal component of all autocorrelation functions. The variance of each parameter was normalized to unity. In this four-dimensional space, we performed spectral clustering[62, 63] (details below) with typically around $K = 10$ clusters. We then identified fast Off cells as the elements of one of the clusters, which was characterized by always being strongly populated as well as by displaying the shortest peak times of the temporal filters, relatively small receptive fields, and a peak in the spike-train autocorrelation at about 10 ms. When there was either apparent contamination of the fast Off cell cluster (identified by variability in the cell parameters and violations of tiling) or if fast Off cells were distributed over two or more clusters (identified by clusters with similar cell parameters and complementary tiling), cells were re-clustered with an adjusted value $K$ for the number of clusters. Note that the aim of the cluster analysis was not to determine the number or identity of all cell types, but simply to extract the single type of fast Off cells.

To perform the spectral clustering, we defined an adjacency matrix $A$, whose elements $A_{ij} = \exp\left( -\left\| \mathbf{X}_i - \mathbf{X}_j \right\|^2 / \sigma^2 \right)$ capture the pairwise Gaussian similarity between cells $i$ and $j$, where $\mathbf{X}_i$ and $\mathbf{X}_j$ are the four-dimensional vectors of the two cells' parameters and $\sigma$ is a scale parameter. We then computed the normalized graph Laplacian matrix[62] $L = D^{-1/2} A D^{-1/2}$, where $D$ is the diagonal matrix with $D_{ii} = \sum_j A_{ij}$, and extracted those $K$ eigenvectors of $L$ that had the largest eigenvalues. In the space that is spanned by these eigenvectors, clustering of data points that are connected by large similarity values is typically simpler than in the original parameter space[62, 63], and we applied k-means clustering to obtain the final set of $K$ clusters. To optimize the scale parameter $\sigma$ for each recording, we used the suggested procedure[62] of performing the clustering for a range of $\sigma$-values (here from 0.1 to 1.0 in steps of 0.005) and selecting the one that yielded the most compact clusters, corresponding to minimal total within-cluster distances of the normalized representations of the $\mathbf{X}_i$ in the space extracted by the eigenvector analysis.

Fast Off cells represented by far the largest homogeneous class of cells in our recordings (comprising up to half of the cells in some recordings). They were also

the only class obtained from the clustering for which we observed receptive field tiling, consistent with previous classification analyses of ganglion cells in the salamander retina[64, 65], where these cells are referred to as "biphasic Off cells". Thus, the present lack of a solid, reliable classification scheme for other ganglion cell types in the salamander retina led us to focus our cell-type analyses on the fast Off and DS cells.

**Spike-triggered non-negative matrix factorization**. For each recorded ganglion cell, we extracted the effective spike-triggered stimulus ensemble from the responses to the spatiotemporal white-noise stimulation in the following way. For each of the $N_{\text{spikes}}$ recorded spikes, we took the preceding 670-ms stimulus sequence and computed a weighted average over time by using the cell's temporal filter as a weight function. Concretely, we regarded the 670-ms stimulus sequence (20 stimulus frames) of contrast values for each pixel as a vector (with values of +1 and −1 corresponding to dark and bright illumination of the pixel, respectively) and computed the scalar product with the corresponding vector from the 670-ms-long temporal filter. Doing this for every pixel yields an effective spatial pattern where each pixel value, which can be positive or negative, measures how well the contrast sequence at the corresponding pixel matched the preferred temporal contrast sequence of the cell. To reduce the analyzed spatial region to the neighborhood of the cell's receptive field, we then only considered the minimal rectangular region in space that fully contained the three-sigma contour of the fitted Gaussian function.

The spike-triggered stimuli form a matrix $S = (s_{ij})$, where the index $i = 1\ldots N_{\text{spikes}}$ counts over all spikes and the index $j = 1\ldots N_{\text{pixels}}$ enumerates all stimulus pixels of the considered rectangular analysis window. We then apply semi-non-negative matrix factorization[36] to the matrix $S$, which aims at identifying a decomposition $W \cdot M$ that best approximates $S$ under the constraint that the elements of the matrix $M$ are all non-negative. Here, $M$ is a $N_{\text{modules}} \times N_{\text{pixels}}$ matrix, whose rows contain the modules of the decomposition, and $W$ is a $N_{\text{spikes}} \times N_{\text{modules}}$ matrix, which contains the weights of each module for each spike-triggered stimulus. Placing a non-negativity constraint only on $M$ allows negative components in $S$ and $W$ so that stimuli can be represented by the positive or negative deviations from the mean light intensity (contrast), which here is the relevant parameter for driving the spiking activity.

In principle, there may be transformations of $W$ and $M$ (for example, linear scaling) that keep $W \cdot M$ fixed and do not affect the non-negativity constraint, which would lead to multiple equivalent decompositions. To reduce the problem of non-unique solutions, we enforced a normalization of the matrix $W$ by constraining each column of $W$ to have unit Euclidean norm, and we applied a common sparsity constraint on the columns of $M$[66, 67]. Together, we thus sought to minimize, under the non-negativity constraint on $M$ and the normalization constraint on $W$, the objective function

$$\| S - W \cdot M \|_{\text{F}}^2 + \lambda \sum_{i=1}^{N_{\text{pixels}}} \| M_i \|_1^2,$$

where the $M_i$ are the columns of $M$, $\| \cdot \|_{\text{F}}$ is the Frobenius matrix norm, $\| \cdot \|_1$ is the $l_1$-norm, and $\lambda$ is a sparseness parameter, which was set to $\lambda = 0.1$ to obtain a reasonable trade-off between sparseness and reconstruction error.

Solutions for this minimization problem can be found iteratively by alternating between optimizing $W$ while keeping $M$ fixed and vice versa. To start the algorithm, the elements of $M$ were initialized with random numbers, drawn uniformly between zero and unity. The optimization of $W$ is a conventional least-squares problem, which can be solved analytically by multiplying $S$ with the pseudoinverse of $M$. This is followed by a normalization of each column of $W$ to unit Euclidean norm. The subsequent optimization of $M$, on the other hand, requires the numerical solution of a non-negative least-squares (NNLS) problem. Fortunately, there are computationally efficient algorithms available[68, 69] that solve the NNLS problem iteratively by systematically probing the space of potential solutions along the boundaries that are defined by the non-negativity constraint (active set method). We here based our implementation of the alternating optimization of $W$ and $M$ on the non-negative matrix factorization Matlab toolbox by Li and Ngom[70].

The alternating optimization of the decomposition is known to rapidly converge to a stationary point where additional iterations do not further change the decomposition matrices[68]. However, there is no guarantee that the obtained stationary point is optimal, that is, that it reflects a global minimum of the objective function. To reduce the impact of local minima of the objective function, we performed the following procedure: We ran the algorithm for a number of iterations, $N_{\text{iter}}$, and then defined the resulting matrix $M$ as the current best solution $M_{\text{best}}$ and assessed its performance by the residual of the reconstruction of $S$. We then inserted a perturbation of $M_{\text{best}}$ (explained below) to obtain a new matrix $M$, which was used as the starting condition for another $N_{\text{iter}}$ iterations. If, after these iterations, the newly resulting $M$ provided an improvement in terms of the residual over $M_{\text{best}}$, $M_{\text{best}}$ was updated to this new solution; otherwise the previous $N_{\text{iter}}$ iterations and the perturbation were discarded. Then a new perturbation of the current $M_{\text{best}}$ was applied, and the process was repeated for a total of $N_{\text{pert}}$ perturbations. In addition, we repeated the entire procedure 100 times with different random initializations of $M$, and the solution with the smallest

residual was selected. We found that, typically, many of the obtained 100 solutions had similarly small residuals and looked very similar or virtually identical, indicating that the chosen procedure successfully avoided erroneous solutions from local minima.

To perform a perturbation of the current best solution, we first identified putative localized subunits by computing for every module (that is, for every row of the matrix $M_{best}$) a measure of the spatial autocorrelation, called Moran's $I$ (see below), and selecting those modules with an autocorrelation value larger than a threshold value of 0.25. We then randomly selected one of the following manipulations, where "noise" always refers to random numbers, drawn uniformly between zero and unity: (1) replacing one randomly selected putative subunit by noise, (2) replacing one randomly selected non-localized module by a duplicate of a randomly selected putative subunit and then adding noise to both copies of the putative subunit, (3) splitting one randomly selected putative subunit into two halves, either vertically or horizontally, with the split occurring next to the pixel with the maximal value of the subunit, and substituting these two new modules for the original putative subunit and a randomly selected non-localized subunit, (4) reinitializing all non-localized modules with noise.

Inserting these perturbations was a crucial ingredient in avoiding local minima of the objective function and obtaining robust solutions. The concrete set of perturbations was chosen heuristically. For applying the STNMF method to different types of data, other types of perturbations may be useful for speeding up the method or for improving robustness. Similarly, the number of considered modules $N_{modules}$, the number of iterations $N_{iter}$ between perturbations, and the total number of performed perturbations $N_{pert}$ are parameters that are likely important for adjusting the method to the specifics of the data, such as the dimensionality of the investigated stimulus space, the expected number of subunits, and the available amount of data. For the present analysis of salamander retinal ganglion cells, we applied $N_{modules} = 20$, $N_{iter} = 20$, and $N_{pert} = 50$, which we found to yield robust results in reasonable computation time. For analyzing data from simulated subunit models, we increased both $N_{iter}$ and $N_{pert}$ to 100, but only used a single run of the algorithm.

**Evaluation of subunits.** To identify subunits from the solution found for the module matrix $M$, we evaluated the localization of each module $m$ (given by each of the rows of $M$, so that for the $k$th module $m$, the elements are given as $m_i = M_{ki}$) as well as its relation to the spiking activity of the neuron. The elements $m_i$, $i = 1, …, N_{pixels}$, of each $m$ correspond to different spatial locations within the analyzed region and can also be viewed as organized in a two-dimensional $N_x \times N_y$ layout with $N_x \cdot N_y = N_{pixels}$. The spatial autocorrelation of each $m$ was calculated as Moran's $I$, defined as: $I = \frac{N_{pixels}}{\sum_i \sum_j w_{ij}} \frac{\sum_i \sum_j w_{ij}(m_i - \overline{m})(m_j - \overline{m})}{\sum_i (m_i - \overline{m})^2}$, where $\overline{m}$ is the mean of the elements of $m$ and the $w_{ij}$ are weights that equal unity whenever the spatial location of $m_i$ is adjacent to that of $m_j$ and zero otherwise. Values of $I$ near zero are found for random distributions over space; localized modules generate values of $I$ larger than zero and bounded by unity.

To relate the modules to the cell's spiking activity, we interpreted each module as a spatial filter and then computed the nonlinearity that relates the activation of this filter to the evoked spike rate, that is, the average number of spikes elicited during the corresponding time bin of the response. To do so, the spatiotemporal white-noise stimulus was again first convolved with the cell's temporal filter. The resulting sequence of spatial stimuli was then filtered by computing the scalar product of the stimulus with the module to obtain a filter output. The filter output was binned into 40 bins in a way so that each bin contained the same number of data points in order to build a histogram of the dependence of the spike rate on the filter output. We displayed the nonlinearity by plotting the average filter output against the corresponding average spike rate for each bin. Finally, we quantified the relation between filter activation and evoked response by computing a gain value from this nonlinearity, which we defined as the difference between the maximal and the minimal spike rate in this histogram. To evaluate the gain of a module independently of the overall excitability of the cell, we also computed in the same way the gain for the spatial receptive field of the cell, obtained from the STA. The ratio of the module gain to the receptive field gain gave us the module's normalized gain.

To automatically detect subunits from among the obtained modules, we applied criteria for both the spatial autocorrelation and the gain of the module. Specifically, we here defined subunits as those modules that had a Moran's $I$ of at least 0.25 or a normalized gain of at least 0.3.

**Analysis of bipolar cell–subunit overlap.** Identified ganglion cell subunits were compared to bipolar cell receptive fields based on the simultaneous recordings of ganglion cells and bipolar cells. We here recorded a total of 20 bipolar cells, three of which were excluded from further analysis because their receptive fields overlapped with two or fewer recorded ganglion cell receptive fields.

To quantify how well the identified ganglion cell subunits matched the receptive field of a simultaneously recorded bipolar cell (Fig. 3), we compared their contours as obtained from the 1.5-sigma outlines of the fitted 2D Gaussians. We compared the area that was shared within the contours of the bipolar cell and of a subunit, $A_{shared}$, to the total area enclosed by the contours, which is given by $A_{total} = A_{BC} + A_{subunit} - A_{shared}$, where $A_{BC}$ and $A_{subunit}$ are the areas of the bipolar cell receptive

field and of the subunit, respectively. The overlap was then computed as the ratio $A_{shared}/A_{total}$. The overlap yields values between zero and unity.

To test whether the encountered overlap values could arise by chance, we obtained the maximal overlap value for each bipolar cell recording and compared the set of these maximal overlap values from the actual data to maximal overlap values from surrogate data. Since the different bipolar cell recordings provided different numbers of recorded ganglion cells with overlapping receptive fields, we first assessed each bipolar cell recording separately by constructing separate surrogate data sets. For each bipolar cell recording, surrogate data were obtained by shuffling and rotating all ganglion cell receptive fields that overlapped with the bipolar cell receptive field (as determined by the 1.5-sigma contours). Concretely, we created for each bipolar cell 1000 surrogate data sets by rotating each ganglion cell receptive field (and thereby the layout of subunits for that ganglion cell) by a random angle between 0 and 360° and by randomly permuting the receptive field positions of the ganglion cells. We then determined the maximal overlap of the bipolar cell receptive field with any of the subunits of each surrogate data set and used the distribution of the 1000 maximal overlap values to determine the probability that a value larger than the observed maximal overlap value would be obtained by chance. Finally, the set of these probability values was tested for significance by Fisher's combined probability test.

**Analysis of subunit overlap in ganglion cell populations.** To quantify how well individual subunits from different ganglion cells matched each other, we computed their relative overlap in the same way as for analyzing the match between subunits and bipolar cell receptive fields. Thus, the overlap is given by $A_{shared}/(A_1 + A_2 - A_{shared})$, where $A_1$ and $A_2$ are the areas within the 1.5-sigma contours of the two subunits and $A_{shared}$ is the shared area.

The analysis of subunit overlaps at the population level (Fig. 5) incorporates data from cells with a wide range of firing rates and noise levels. To, nonetheless, make this analysis robust against the occurrence of spurious subunits, we here included only subunits that robustly occurred in the analysis for different random initializations of the module matrix $M$. Concretely, we used Gaussian fits to compute the center points for all final subunits from all 100 runs with different initializations (see section on "Spike-triggered non-negative matrix factorization"). We then took the subunits from the single best run and checked for each of these whether within a circle of 30-μm radius around its center point there were subunit center points in at least 50 of the 100 runs. If that was the case, we defined a robust subunit as the pixel-wise average of all the subunits with center points in this circle.

To compare the distribution of overlap values with chance level, we randomly permuted the receptive field center positions of the ganglion cells within each cell type and recording and then computed the overlap values between the correspondingly relocated subunits. This shuffling procedure was repeated 100 times to obtain mean values and standard deviations for the distribution of overlap values (Fig. 5).

**Response predictions.** To assess the relevance of the subunit layout for predicting ganglion cell responses, we compared a subunit model and a standard LN model for predicting responses to three different stimulus sets: (1) held-out spatio-temporal white-noise sequences, which were repeatedly inserted as identical stimulus sequences (frozen noise) between longer, non-repeated white-noise segments; (2) a set of 300 briefly flashed natural images; and (3) shifted natural images that used brief flashes of the same image with slightly different spatial positioning (see "Visual stimulation"). For these model evaluations, the model parameters were generally obtained from the non-repeated white-noise segments so that the evaluations were performed on stimuli that were not used for obtaining parameters. Only in the analysis of natural images at slightly different positions, one parameter (the gain of the final output nonlinearity in each model) remained free and was fitted to the test data.

The spatiotemporal white-noise sequences were first convolved with the cell's temporal filter to yield a sequence of effective spatial images. For all stimuli, model predictions were constructed by first computing a filter signal, based on either the cell's spatial receptive field (LN model) or based on the set of identified subunits (subunit model). For the LN model, the filter signal $F_{LN}$ was simply obtained by filtering the images with the spatial receptive field, as retrieved from the spike-triggered average. For the subunit model, the images were first filtered by each subunit individually. These filter outputs were half-wave rectified and then summed in a weighted manner in order to obtain the filter signal $F_{subunit}$. Note that this half-wave rectification is likely not the correct or best subunit nonlinearity and was taken here for simplicity and for balancing the number of parameters in the different models, making the evaluation of the subunit model a conservative estimate of its ability to predict responses. Since our primary goal here was to evaluate whether using subunits can improve response predictions over linear spatial summation as in the LN model and not to optimize response predictions, we did not try fitting subunit nonlinearities to the data, but rather aimed at minimizing the number of model parameters. The weights for the summation of subunit signals were determined from a least-squares fit of the receptive field by the subunits, so that individual image pixels had similar relevance for the LN model prediction and the subunit model prediction.

To turn $F_{LN}$ and $F_{subunit}$ into predictions for the frozen white-noise sections, we computed nonlinearities for both models from the non-repeated parts of the white-noise stimulus. This was done by relating $F_{LN}$ and $F_{subunit}$ to the average

evoked firing rate in a histogram manner, similar to the computation of the gain for the subunits (see "Evaluation of subunits"), using again 40 bins of the filter signals and computing the average filter signal and average firing rate for each bin. The histograms were then fitted with nonlinear functions of the form $r(F) = a_1 \cdot \ln\left(1 + e^{a_2 \cdot (F + a_3)}\right)$ by optimizing the parameters $a_1$, $a_2$, and $a_3$ according to a least-squares criterion. The fitted functions were used to obtain response predictions for the frozen-noise sections. To quantify model performance, we computed for each model the correlation coefficient $R$ between prediction and measured firing rate and reported the explained variance $R^2$.

The models can only be expected to explain such variations of the firing rate that correspond to a deterministic signal, whereas any noise in the estimated firing rates provides variance that is inaccessible ("unexplainable") to the model. We checked to what extent noise in the estimated ganglion cell firing rate might limit the explained variance by computing the explainable variance[26] for each cell. Concretely, we separated the responses to the frozen noise into even and odd trials, computed the firing rate profiles for each, and calculated the explained variance $R^2$ between them[26]. This is a measure of the reliability of the structure in the PSTH, that is, a measure of the explainable variance, which a model may hope to capture. We found that this measure was generally close to unity ($0.97 \pm 0.04$, mean $\pm$ SD, $N = 28$ cells), indicating that noise in the estimation of the ganglion cell firing rates is not a limiting factor here, as expected from the large number of repeats of the frozen noise (more than 200).

For the set of 300 natural images, we did not use a nonlinearity to turn $F_{LN}$ and $F_{subunit}$ into actual response predictions for the evoked spike count, but rather, we used $F_{LN}$ and $F_{subunit}$ directly to predict the rank order of the images in terms of their average spike count. This procedure just assumes that any nonlinear transformation of the filter signals is monotonic, yielding larger spike counts for larger filter signals. Not applying an output nonlinearity makes the model evaluation independent of the accuracy of estimating the nonlinearity, which is likely to be different for the image presentation as compared to the white-noise stimulation because of the different stimulus dynamics and because of adaptation. Thus, we quantified model performance by computing Spearman's rank correlation coefficient for the relation between the image order according to the model prediction and the image order according to the measured average spike counts.

For the shifted natural images, presented at nine different positions, we used a different approach because using only nine positions was insufficient to get good sampling for the rank correlation. Instead, we here applied a half-wave rectification to the filter signals $F_{LN}$ and $F_{subunit}$ in order to predict the actual spike count elicited by the nine positions. The slope of the linear part of the half-wave rectification was left as a free parameter for each model, which was optimized by minimizing the root-mean-square (RMS) error. Model performance was then quantified for each cell and each image by the obtained RMS error values.

For comparison, we also assessed performance of a model where the subunits were scrambled (shuffled subunit model) by randomly permuting the pixel values between the different subunits. This shuffling was done for each image pixel separately so that each pixel was filtered with the same set of weights as for the original subunits, but in different compositions. After this subunit shuffling, the corresponding model was obtained and evaluated in the same way as the original subunit model.

For comparison with the NIM[28] (Supplementary Fig. 9), we analyzed the ganglion cells recorded under spatiotemporal white noise. We based our analysis on the NIM Matlab toolbox (http://neurotheory.umd.edu/nimcode). However, since it was not practically possible to fit the full version of this model for the long recording durations and the high-dimensional spatiotemporal stimuli of our data, we simplified the model fitting as follows: as for the application of the STNMF method, we integrated out time by convolving the stimulus sequence with the temporal filter of the analyzed cell, and we set the post-spike filter in the model to zero. Thus, the only components to be optimized were the spatial subunits and their nonlinearities. We initialized the model with normally distributed random numbers for the subunits and threshold-linear subunit nonlinearities. (We also tested exponential subunits with no difference in results.) The number of considered subunits was set for each cell to the number of subunits identified by the STNMF method. These simplifications allowed us to apply the parameter fitting of the NIM for the large stimulus space considered in our analysis and to directly compare model performance, as temporal binning remained constant for the different models. The parameters for model optimization and regularization were kept identical to the original NIM implementation. Yet, clearly, our analysis of this reduced and restricted model does not aim at a full evaluation of the NIM's potential to identify subunits or predict ganglion cell responses.

**Code availability**. The computer code (in Matlab) used for applying the STNMF method to experimental as well as simulated data is available from https://github.com/gollischlab/STNMFanalysis.

**Data availability**. All relevant data are available from the authors upon request.

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

## Acknowledgements

This work was supported by the European Union Seventh Framework Programme (FP7-ICT-2011.9.11) under grant agreement number 600954 (VISUALISE), by the Horizon 2020 Programme (H2020-MSCA-IF-2014) under grant agreement number 659227 (STOMMAC), by the Deutsche Forschungsgemeinschaft (GO 1408/2-1 and Collaborative Research Center 889, C1). and by the European Research Council (ERC) under the European Union's Horizon 2020 research and innovation programme (grant agreement number 724822). We furthermore acknowledge support by the Open Access Publication Funds of Göttingen University.

## Author contributions

J.K.L. and T.G. designed the experiments and data analysis. J.K.L. conducted the ganglion cell recordings from salamander retina and performed the subunit analyses of recorded ganglion cells. H.M.S. conducted the combined recordings of bipolar cells and ganglion cells and performed analyses and imaging of bipolar cells. V.K. conducted ganglion cell recordings from mouse retina. M.H.K. assisted in the bipolar cell recordings and in data analysis. A.O., S.P., and T.G. provided computational tools and assisted in data analysis. J.K.L., A.O., and T.G. designed and analyzed the computational models. F.R. developed and applied the clustering analysis of ganglion cell types. J.K.L. and T.G. wrote the paper with inputs from all authors.
