## [Peer Review File · Nature Communications]

Reviewers' Comments:

Reviewer #2 (Remarks to the Author):

This manuscript reports how to automatically discover subunit structure in the response properties of retinal ganglion cells by analyzing the statistical structure of spike-evoking stimuli drawn from a white-noise stimulus set. The authors perform the analysis using nonnegative matrix factorization with nonnegative and sparsity constraints on the subunit field values. They then go on to show that the discovered units correlate with the receptive fields of experimentally measured bipolar cells, and that the responses from the subunit model are significantly better than the standard LN model for other stimuli.

I had some questions concerning the analysis. First, it wasn't exactly clear how the STA stimuli were processed. The authors state that SVD was used to first decompose the spatiotemporal patterns into a spatial filter and temporal filter, presumably using the top rank-1 decomposition. Then they say that a weighted average of the spatial patterns is performed, weighted by the temporal filter from the SVD. However, the temporal filter contains both positive and negative values, so it is not clear how the weighted average is performed. In particular, if the sum of the weights is near zero, I would imagine the normalization would be extremely noisy.

I would have also liked to have seen if more traditional second order analysis of the stimuli would yield meaningful results. For example, a spike-triggered covariance analysis could have been performed, and the leading eigencomponents could have been compared to the results of STNMF. It probably is the case that this analysis would result in non-localized fields, but it would still be instructive to compare this spike-triggered Gaussian-based analysis with STNMF.

Also, how did the authors utilize the weights from the NMF analysis? Are those weights used for the subunit model, or were the weights in the model after the rectification nonlinearity fixed or fit from scratch? It would be nice to ascribe some meaning to the mixing weights of the subunits in the analysis.

I appreciate that the subunit model seems to roughly describe the ganglion responses on novel natural image stimuli. Another quantitative measure of the performance of the model would have been to see how well it performs on a held-out test set from the white-noise stimuli patterns. It would be instructive to see how well the subunit model performs relative to the LN model, or Gaussian model on such a test set.

Reviewer #3 (Remarks to the Author):

In their paper, Liu et al. propose a new method based on non-negative matrix factorization to decompose the receptive field of a neuron into subunits, which are nonlinearly integrated. They demonstrate the power of their approach by applying it to recordings from salamander retinal ganglion cells, where the identified subunits correspond to individual bipolar cells, as impressively demonstrated through paired recordings from ganglion and bipolar cells. Overall, this is an interesting contribution with a potentially clear methodological advance; the new method may help to bring functional models of neurons (such as the LN model) and anatomy closer together and thus has huge potential in systems neuroscience. The statistical methods used are adequate and rigorous (for a few suggestions, see below) and the description of the method is provided at sufficient level of detail (yet, code would be helpful, see below).

While of potential wide applicability and interest, the paper lacks in certain aspects in its present form. As this is clearly a methodological paper, I would like to see more depth regarding (a) evaluation of the robustness of the proposed method, (b) evaluation of the generality of the method, (c) comparison with other methods and (d) discussion of potential new findings that can only be achieved with this method.

(a) The authors evaluate in Fig. 2 and SFig. 3 some general properties of their model and show that it can recover subunits if present in the data and is robust against variations in the temporal kernels. I would like to see more exploration of the robustness of the model: How many minutes of recordings/spikes does one need to robustly extract subunits? What happens if model assumptions are violated, e.g. the true subunits are quadratically combined like in STC? Was that ever observed in the data? What is the effect of noise level in toy data?

(b) While it is impressive to see that the recovered subunits in the salamander in some cases correspond to receptive fields of bipolar cells, it would be nice to see how this method performs with data from other species, such as mice. I realize that this may be asking too much, but showing subunits of mouse RGCs would be nice, even without the dual recordings, if the authors have the data.

(c) The authors show that their method performs better than the LN model in predicting RGC spikes. But the LN model is somewhat of a straw man – there is a host of more complex models around and it would be instructive to compare the prediction performance to those. At the very least models like that of McFarland et al. (2013, ref. 28) and a few other state of the art techniques (e.g. Freeman et al. eLife, 2015, ref 29; Theis et al. Plos Comp Bio 2013) with a similar spirit should be compared. This would also be a chance to clarify whether the authors expect this model to yield good prediction performance or whether the advantage of the model is rather its interpretability. Also the comparison should include an estimate of the explainable variance based on the oracle prediction on the frozen noise, see e.g. ref 26.

(d) The Discussion is a bit thin regarding potential applications – where do the authors think this method may be more useful than other methods or enable testing biological theories not otherwise testable? I would like to see the authors describing potential applications much more explicitly.

Other comments

- Shuffle tests: From the main text, it remains often unclear what exactly was shuffled to test which hypothesis (Fig. 4 and 5). Please make this more accessible and transparent.
- Being a methods paper, it could use a little bit more mathematical details/rigor in the main text. I sympathize with the authors trying to make the work accessible to non-experts, but the descriptions in the main text are often too vague.
- The methods are not explicit about which data the models were fitted and evaluated on for the model comparison, i.e. whether the evaluation was performed on a real test set.
- There is a note in the methods (p15) about rectifying subunit nonlinearities not being the best. Why not replace them by something simple (e.g. piecewise linear)?
- The discussion of the literature with regards to previous papers on subunit models is a bit biased. I think Freeman et al. have nicely demonstrated that very likely their extracted subunits correspond to BCs sampling from distinct sets of photoreceptors. Also labeling this a method with “no prior assumptions” (p. 2) is a bit strong (of course there are inherent assumptions, see comment about quadratic features above).
- What is meant by “we quantified the maximal difference in firing rate along this curve” on p. 3?
- Are there any theoretical considerations why the gain, the nmf weight and the Rf fit should be similar?
- I would like to see the examples of where there wasn't a match of any inferred subunit to a patched BC on p 4/ Fig 3.
- Fig. 3, BC image: lacks scale bars, any way to reduce blur by deconvolution?
- Fig 3b: What is meant precisely by ganglion cell receptive field?
- Fig. 4b: Given the spikes predicted by the subunit model, I find it hard to believe the better numbers, the prediction seems not really better than the prediction of the LN. Maybe the authors could show additional examples in the supplement and add rate estimates?
- Fig. 4: For the VE and rank correlation plots, histograms of pairwise differences would also be useful.
- The evidence for subunit sharing I don't find particularly convincing in Fig. 5 – I think the authors argue that the histogram is a bit higher than their shuffle curve in the tails... but the authors also show 1.5 sigma outlines of the shuffle results, which is not exactly standard (i.e. the green lines theoretically contain 86% of the shuffle runs).

- Is it surprising that functionally different RGCs like in Fig. 5 share BCs in light of the connectomics data (Helmstaedter 2013 and others; p. 6)?
- A $DSI > .25$ seems a rather loose criterion for establishing DS. Why not perform a significance test instead? Also are there not multiple DS types in the salamander?
- How do the subunit mosaics look for other cell types?
- Figures generally: More subpanel labels would help.
- The paper would benefit tremendously from a software package with data that allows easy replication of the findings as well as using it on ones own data.

We thank the Reviewers for their helpful and constructive comments. As detailed below, we have addressed all points raised by the Reviewers. Most importantly, we added new analyses of comparison with alternative methods aimed at resolving the subunit structure of receptive fields and of how robust the STNMF method is with respect to the shape of subunit nonlinearities, to noise, and to the number of available spikes. Furthermore, we performed additional experiments to show that the method is also applicable to mouse retina. Finally, we thoroughly revised the text in order to provide a more rigorous as well as more intuitive access to the presented method. We believe that the manuscript has strongly benefited from these changes.

Reviewer #2 (Remarks to the Author):

This manuscript reports how to automatically discover subunit structure in the response properties of retinal ganglion cells by analyzing the statistical structure of spike-evoking stimuli drawn from a white-noise stimulus set. The authors perform the analysis using nonnegative matrix factorization with nonnegative and sparsity constraints on the subunit field values. They then go on to show that the discovered units correlate with the receptive fields of experimentally measured bipolar cells, and that the responses from the subunit model are significantly better than the standard LN model for other stimuli.

I had some questions concerning the analysis. First, it wasn't exactly clear how the STA stimuli were processed. The authors state that SVD was used to first decompose the spatiotemporal patterns into a spatial filter and temporal filter, presumably using the top rank-1 decomposition. Then they say that a weighted average of the spatial patterns is performed, weighted by the temporal filter from the SVD. However, the temporal filter contains both positive and negative values, so it is not clear how the weighted average is performed. In particular, if the sum of the weights is near zero, I would imagine the normalization would be extremely noisy.

Indeed, we used to top-rank components from the SVD to obtain the temporal filter and spatial receptive field. We've added a corresponding note in the Methods section (page 14 under "Receptive field analysis"). The obtained temporal filter can have positive and negative values. For normalization of the temporal filter, the Euclidean norm (L-2 norm) is used so that the sum of squares of the filter elements equals unity after normalization, not the integral. Thus, the normalization is robust even if the sum of filter elements is near zero. We have revised the explanation of the normalization to make this clearer (page 14). Note that the normalization of the filter is performed before it is applied to compute the weighted average of stimuli. This weighted average is essentially a scalar product between the filter and a temporal sequence of contrast values for a given image pixel. Both the filter and the stimulus sequence typically contain positive and negative values, and the scalar product can be positive or negative. Values near zero here mean that the considered pixel is expected to neither enhance nor suppress activity of the considered ganglion cell. In the revision, we have extended the explanation of the temporal weighting along these lines (page 3 and page 16).

I would have also have liked to have seen if more traditional second order analysis of the stimuli would yield meaningful results. For example, a spike-triggered covariance analysis could have been performed, and the leading eigenvectors could have been compared to the results of STNMF. It probably is the case that this analysis would result in non-localized fields, but it would still be instructive to compare this spike-triggered Gaussian-based analysis with STNMF.

We thank the Reviewer for this suggestion. Indeed, it is instructive to compare our results with a spike-triggered covariance (STC) analysis. As suggested, we have performed STC analysis to extract sets of spatial filters for comparison with the subunits obtained by our STNMF method. The results are shown in a new supplementary figure (Suppl. Fig. 3) and discussed in the text (page 4f.). The analysis indicates that, for the purpose of extracting physiologically relevant subunits, the STNMF method is superior to the STC method in two respects: First, STC analysis is more sensitive to the high dimensionality of the analyzed stimulus space (which must include many pixels in order to provide for sufficient spatial resolution) so that often relevant stimulus features do not stand out by their eigenvalues from the broad distribution of non-relevant eigenvalues and the extracted features are correspondingly noisy. Second, when features are extracted from the STC analysis, they are typically, as expected by the Reviewer, non-localized, rather corresponding to Fourier modes of the receptive field than to potential presynaptic circuit elements. The comparison thus underscores the power of the STNMF analysis for the specific purpose of extracting localized subunits as candidates for presynaptic receptive fields.

Also, how did the authors utilize the weights from the NMF analysis? Are those weights used for the subunit model, or were the weights in the model after the rectification nonlinearity fixed or fit from scratch? It would be nice to ascribe some meaning to the mixing weights of the subunits in the analysis.

The weights obtained from the NMF analysis are not the weights that we apply in the subunit model. In this model, subunit weights were obtained from the parameters needed to fit the receptive field by a linear combination of the subunits. This was done to ensure that the overall contribution of each pixel best matched its contribution in the standard LN model. As shown in Supplementary Fig. 2, however, these receptive field fitting weights are similar to the average weights from the NMF analysis (“mixing weights”), which could thus here have equally been used as weights for the subunit model. We have expanded the discussion of these different weight measures to clarify their relation (page 4). The individual mixing weights of the NMF analysis only signify how much a given subunit contributed to the generation of a given spike.

I appreciate that the subunit model seems to roughly describe the ganglion responses on novel natural image stimuli. Another quantitative measure of the performance of the model would have been to see how well it performs on a held-out test set from the white-noise stimuli patterns. It would be instructive to see how well the subunit model performs relative to the LN model, or Gaussian model on such a test set.

The results of Fig. 4b are, in fact, from an analysis of held-out stimulus segments from the spatiotemporal white-noise experiments. The results show that the subunit model outperforms the LN model in a similar fashion as for the natural images (Fig. 4c). We have revised the text to more clearly point to this analysis of held-out stimulus segments (page 7 and page 20).

Reviewer #3 (Remarks to the Author):

In their paper, Liu et al. propose a new method based on non-negative matrix factorization to decompose the receptive field of a neuron into subunits, which are nonlinearly integrated. They demonstrate the power of their approach by applying it to recordings from salamander retinal ganglion cells, where the identified subunits correspond to individual bipolar cells, as impressively demonstrated through paired recordings from ganglion and bipolar cells. Overall, this is an interesting contribution with a potentially clear methodological advance;

the new method may help to bring functional models of neurons (such as the LN model) and anatomy closer together and thus has huge potential in systems neuroscience. The statistical methods used are adequate and rigorous (for a few suggestions, see below) and the description of the method is provided at sufficient level of detail (yet, code would be helpful, see below).

While of potential wide applicability and interest, the paper lacks in certain aspects in its present form. As this is clearly a methodological paper, I would like to see more depth regarding (a) evaluation of the robustness of the proposed method, (b) evaluation of the generality of the method, (c) comparison with other methods and (c) discussion of potential new findings that can only be achieved with this method.

(a) The authors evaluate in Fig. 2 and SFig. 3 some general properties of their model and show that it can recover subunits if present in the data and is robust against variations in the temporal kernels. I would like to see more exploration of the robustness of the model: How many minutes of recordings/spikes does one need to robustly extract subunits? What happens if model assumptions are violated, e.g. the true subunits are quadratic ally combined like in STC? Was that ever observed in the data? What is the effect of noise level in toy data?

We agree that an analysis of robustness of the model is an important addition. In the revision, we have tackled this issue in the following ways. First, we have checked that the assumed monotonic, threshold-like shape of the subunit nonlinearity is not a prerequisite for the extraction of subunits. For a model with quadratic subunit nonlinearities, as suggested by the Reviewer, the method works equally well; see the new Supplementary Fig. 4. (But no, we have not encountered any symmetric subunit nonlinearities in our data.) Note that, in this process, we have also extended the analysis of the model with different temporal filters for different subunits to now contain five overlapping subunits (Suppl. Fig. 5), and we corrected the display of the original model (Fig. 2) to show the threshold-quadratic subunit nonlinearities that we had used here. (Although the method also works, of course, with the threshold-linear subunit nonlinearities that had originally been displayed.)

Second, we used model simulations to check how the extraction of subunits depends on the number of spikes used for the analysis and on the level of added noise. These analyses provided two important insights. First, the subunit estimation is relatively robust to noise and spike deletions up to a certain point after which performance drops steeply. Second, when subunit detection starts to fail, it does so primarily by losing individual subunits to noise while other subunits are still faithfully retained. We have checked that a similar dependence on noise and spike numbers also holds for actual experimental data. The results of this robustness analysis with respect to noise and spike number are shown in the new Supplementary Fig. 6 and discussed in the text (page 5f.).

(b) While it is impressive to see that the recovered subunits in the salamander in some cases correspond to receptive fields of bipolar cells, it would be nice to see how this method performs with data from other species, such as mice. I realize that this may be asking too much, but showing subunits of mouse RGCs would be nice, even without the dual recordings, if the authors have the data.

We agree that showing data from a mammalian retina could strengthen the presentation of STNMF as a generally applicable method. We therefore performed recordings from mouse retina under spatiotemporal white noise at high spatial resolution (pixel size down to 15 μm). We find that STNMF yields localized subunits in a very similar fashion as for the salamander

data. This underscores that STNMF is readily applicable to other systems. We have included a new supplementary figure (Suppl. Fig. 10) to show data from mouse retina.

(c) The authors show that their method performs better than the LN model in predicting RGC spikes. But the LN model is somewhat of a straw man – there is a host of more complex models around and it would be instructive to compare the prediction performance to those. At the very least models like that of McFarland et al. (2013, ref. 28) and a few other state of the art techniques (e.g. Freeman et al. eLife, 2015, ref 29; Theis et al. Plos Comp Bio 2013) with a similar spirit should be compared. This would also be a chance clarify whether the authors expect this model to yield good prediction performance or whether the advantage of the model is rather its interpretability. Also the comparison should include an estimate of the explainable variance based on the oracle prediction on the frozen noise, see e.g. ref 26.

We now clarify in the text that the main purpose of the present work is to extract physiologically interpretable subunits, rather than to optimize response predictions (page 8). Nonetheless, as suggested, we now relate our method of identifying subunits to alternative methods. First, we explored how STNMF compares to spike-triggered covariance analysis in terms of identifying localized subunits (new Suppl. Fig. 3; see also response to comment by Reviewer #2). Furthermore, we also added a comparison of our method and the corresponding model predictions to the Nonlinear Input Model (NIM) introduced by McFarland et al. (2013). The results are shown in a new supplementary figure (Suppl. Fig. 9) and discussed in the text at the end of the section on response predictions (page 8).

We also looked at the other suggested methods by Freeman et al. and Theis et al. We found, however, that adjusting these different methods to our data is not as straightforward as it may seem at first sight, primarily because of the high dimensionality of the relevant stimulus space. Note that our experiments applied (by necessity) spatiotemporal white-noise stimuli with a much higher spatial resolution than typically used in receptive field analyses, yielding a stimulus space of typically several hundred dimensions (considering only spatial stimulus components). We found that – at least in our hands – the suggested alternative methods struggled with this high dimensionality both in terms of computer runtime and of getting stuck in local minima. For example, fitting a full NIM to the data of a single ganglion cell from our recorded data set had still not converged after more than 30 days of runtime on a 12-core/128-GB-RAM machine. Thus, making these models fully compatible with the data at hand would require adjustments of modeling parameters and exploration of fitting strategies that we consider well beyond the scope of this work.

We therefore settled for an investigation of a reduced version of the NIM. We reduced model complexity by integrating out time (as we did for our STNMF analysis) and by not considering any post-spike filter. This furthermore allowed us to directly compare model performance without the need to resample time. For this reduced NIM, we found that the obtained filters do not show the localized structure observed for the subunits of the STNMF method and that it does not yield better response predictions than the STNMF-derived subunit model. With respect to these results, we added a cautionary note that our approach certainly did not explore the full potential of this model (page 8). Nonetheless, these investigations underscore the merit of methods that do not require fitting of entire input-output models.

Finally, we estimated the explainable variance for each cell in the analysis of response predictions on frozen noise by separating the data into even and odd trials and computing the variance explained between them. As expected, because of the large number of trials that went into computing the PSTHs (more than 200 trials), the explainable variance is very close

to unity (0.97 ± 0.04 , mean \pm SD). Thus, noise in the PSTHs does not limit the measure of variance explained. We therefore kept the measure of variance explained in the figures, but now explain the considerations and results regarding the explainable variance in the Methods section under “Response predictions” (page 21).

(d) The Discussion is a bit thin regarding potential applications – where do the authors think this method may be more useful than other methods or enable testing biological theories not otherwise testable? I would like to see the authors describing potential applications much more explicitly.

As suggested, we have added a new paragraph (at the end of the Discussion; page 11) where we highlight potential applications of the method.

Other comments

- Shuffle tests: From the main text, it remains often unclear what exactly was shuffled to test which hypothesis (Fig. 4 and 5). Please make this more accessible and transparent.

Agreed. We have reworked these text passages and expanded the explanations around the shuffling analyses (page 7 regarding the shuffle analysis of bipolar cell/subunit overlap and page 9 regarding the shuffle analysis of subunit overlap for populations of ganglion cells).

- Being a methods paper, it could use a little bit more mathematical details/rigor in the main text. I sympathize with the authors trying to make the work accessible to non-experts, but the descriptions in the main text are often too vague.

We have substantially reworked the presentation of the method in the main text, aiming at providing more rigor and conceptual details, yet still trying to keep it generally accessible by referring to the Methods section for technical details and formulas.

- The methods are not explicit about which data the models were fitted and evaluated on for the model comparison, i.e whether the evaluation was performed on a real test set.

We have now clarified, both in the main text (page 7) and in the Methods section (page 20), that the model evaluations are performed on real test sets. The model parameters (i.e., receptive field, subunits, nonlinearities) were obtained from the non-repeated segments of the spatiotemporal white-noise experiments. For the predictions in response to spatiotemporal white noise, the evaluation was performed on inserted, repeated frozen-noise segments, which thus represent held-out data. For the predictions to natural images, the images were only used for model evaluations; the model parameters are again taken from the non-repeated white-noise segments. The only exception was the gain of the final output nonlinearity for the analysis of natural images at slightly different position (Suppl. Fig. 8), which was fitted for each analyzed model to the test data.

- There is a note in the methods (p15) about rectifying subunit nonlinearities not being the best. Why not replace them by something simple (e.g. piecewise linear)?

We now more clearly state that the main interest in the models in this work is to show that the STNMF method extracts meaningful and useful subunit layouts, not to optimize response predictions (page 8). The simple rectifying subunit nonlinearities already allow us to see a strong improvement over the LN model without introducing further free parameters to model the nonlinearities. We tried out different simple subunit nonlinearities, such as threshold-

quadratic and exponential, but did not find substantial differences in model performance. Note that parameterizing the subunit nonlinearities would require a simultaneous optimization with the output nonlinearity, which would introduce considerable additional complexity. Thus, although we agree that trying to optimize the subunit nonlinearities will be an interesting direction, we feel that this is beyond the scope of this work and might rather distract from the main point. We have aimed at better clarifying the purpose of using a fixed subunit nonlinearity in the Methods section (page 20).

- The discussion of the literature with regards to previous papers on subunit models is a bit biased. I think Freeman et al. have nicely demonstrated that very likely their extracted subunits correspond to BCs sampling from distinct sets of photoreceptors. Also labeling this a method with “no prior assumptions” (p. 2) is a bit strong (of course there are inherent assumptions, see comment about quadratic features above).

Yes, agreed. We have rephrased the statement in the introduction, specifying that connecting inferred subunits to circuit elements is a general challenge and pointing out the progress made by Freeman et al. (page 2). Regarding the second point, we did not mean to imply that the STNMF method contains no prior assumptions, but rather that methods without prior assumptions are desirable and that the STNMF method does not require a prior constraints on the spatial compactness of subunits. However, we agree that the formulation was misleading, and we have changed it to say that methods with “minimal” prior assumptions are desirable and that the STNMF method does not require explicit prior specification of subunit size, shape, number, or nonlinearity (page 2).

- What is meant by “we quantified the maximal difference in firing rate along this curve” on p. 3?

We apologize for the lack of clarity here. We rephrased and expanded this (now page 4) to clarify that we simply took the histogram of filter outputs versus firing rates and computed the difference between the maximum and minimum of the histogram.

- Are there any theoretical considerations why the gain, the nmf weight and the Rf fit should be similar?

The consideration behind this is that these three measures all capture how strong the influence of a subunit on the spiking activity of the ganglion cell is, that is, what the connection strength is. The fact that they are similar (apart from a scaling factor) corroborates this interpretation, though we do not have a formal reason of whether they should be linearly related to each other. In the revised text, we have aimed at clarifying our interpretation of these measures (page 4).

- I would like to see the examples of where there wasn't a match of any inferred subunit to a patched BC on p 4/Fig 3.

We now include a new supplementary figure (Suppl. Fig. 7) with more examples of recorded bipolar cells and subunit layouts, some with good matches, others where there was no match found. Also note that we have meanwhile increased the number of recorded bipolar cells to 17 and have adjusted the text accordingly (page 6).

- Fig. 3, BC image: lacks scale bars, any way to reduce blur by deconvolution?

We added a scale bar and performed deconvolution of the image, based on a measurement of the point-spread function by imaging fluorescent beads. Note, however, that image quality is compromised by the fact that images were obtained from retinas that, after several hours of recording, had to be peeled off a perforated MEA, to which it had been tightly attached by suction. Yet, the characteristic morphology of a bipolar cell is clearly discernable.

- Fig 3b: What is meant precisely by ganglion cell receptive field?

The receptive field here refers to the spatial component of the spike-triggered average under spatiotemporal white noise, represented in a pixel-by-pixel fashion. We have clarified this now in the legend of Fig. 3.

- Fig. 4b: Given the spikes predicted by the subunit model, I find it hard to believe the better numbers, the prediction seems not really better than the prediction of the LN. Maybe the authors could show additional examples in the supplement and add rate estimates?

Yes, the example in the previous manuscript version was not really representative of the differences we observe between the subunit model and the LN model and was thus not well chosen. We have replaced this now with a more typical example, which also better reflects that some of the spiking events in the data are reproduced by the subunit model, but not by the LN model.

- Fig. 4: For the VE and rank correlation plots, histograms of pairwise differences would also be useful.

Histograms of pairwise differences have been added to the plots.

- The evidence for subunit sharing I don't find particularly convincing in Fig. 5 – I think the authors argue that the histogram is a bit higher than their shuffle curve in the tails... but the authors also show 1.5 sigma outlines of the shuffle results, which is not exactly standard (i.e. the green lines theoretically contain 86% of the shuffle runs).

The excess of large overlaps in the data compared to shuffled receptive fields may not look impressive. However, this impression partly comes from the large numbers in the histogram from small overlap values, which naturally occur much more frequently. Note that, for example regarding the fast-Off-versus-DS-cell data, there are several tens of subunit pairs with strong overlap (relative overlap > 0.5), amounting to about threefold the number expected by chance. In the revised manuscript, we have expanded the discussion of these results and added quantification of the number of strongly overlapping subunits to more clearly point out the conclusions (page 9). Also, please note that the green shaded region of the shuffle data correspond to the one-sigma region. The 1.5-sigma value referred to the size of the subunits that was used for the overlap calculation, which is now stated in the main text (page 7).

- Is it surprising that functionally different RGCs like in Fig. 5 share BCs in light of the connectomics data (Helmstaedter 2013 and others; p. 6)?

Indeed, sharing of BC inputs by functionally distinct RGCs can be expected based on anatomical grounds, and Helmstaedter et al. (2013) provide the nice example of local edge detectors and ON-OFF DS cells, which both appear to be strongly connected to bipolar cells of the type called CBC5R. We now point out this relation to anatomical data and clarify that

the analysis of shared subunits provides for a functional complement to these studies (now moved to the Discussion, page 10).

- A $DSI > .25$ seems a rather loose criterion for establishing DS. Why not perform a significance test instead? Also are there not multiple DS types in the salamander?

Regarding the chosen threshold of the DSI, please note that we define the DSI via the vector sum of the firing rates, not the difference between preferred and null direction. While the latter might be more commonly used, the former yields a more robust measure of DS (at least in our hands), but also typically produces smaller values. (See, e.g., Rivlin-Etzion, Wei, Feller, *Neuron* 2012, where a threshold of 0.3 is used for the index obtained from the preferred minus null response, but a threshold of 0.2 for the vector sum). Yet, we agree that including a significance analysis may improve the detection of DS cells, and we now include only those cells that have a $DSI > 0.25$ and a significant tuning as determined by a shuffle analysis. (Basing DS cell detection only on significance analysis also appeared too loose a criterion as it yielded cells with very weak tuning as potentially DS.) The significance analysis is now explained in the Methods section (page 14). The effects of including the significance analysis on the population analysis of Fig. 5 were minor; only three cells had to be discarded from the group of DS cells.

Regarding multiple DS types in salamander, indeed, we recently showed that DS cells exist in salamander and that one can distinguish two types (Kühn, Gollisch, *J Neurosci* 2016). We now clarify this in the Methods section and explain that the current set of DS cells likely contains primarily one of the two types, though not exclusively (page 14; see also new note in Results about this on page 8).

- How do the subunit mosaics look for other cell types?

Our clustering did not aim at separating out different cell types, but focused on extracting the particular cell type of “fast Off” cells. This group is by far the most homogeneous and most commonly encountered in our recordings, and the corresponding cluster is the only one that shows clear tiling at present. This is consistent with previous clustering analyses of salamander ganglion cells from the group of Michael Berry. Other clusters in our analyses most likely do not represent specific ganglion cell types, and we therefore do not consider them further at this point. We added explanations along these lines in the Methods section under “Cell-type classification” (page 15).

- Figures generally: More subpanel labels would help.

As suggested, we have added more subpanel labels in several of the figures.

- The paper would benefit tremendously from a software package with data that allows easy replication of the findings as well as using it on ones own data.

Agreed. We have made our code for applying the STNMF method available via a public repository and added the link to the manuscript (page 22). In addition, we provide sample data and explanatory notes in the repository, aimed at helping users apply the methods to their own data.

Reviewers' Comments:

Reviewer #2 (Remarks to the Author):

I am satisfied with the revised manuscript containing clarifications and new analyses that address my previous concerns.

Reviewer #3 (Remarks to the Author):

The authors have addressed all my comments. I think the paper will have a big impact on sensory neuroscience given the rising interest in linking structure of neural circuits and function.

The only remaining suggestion I have is that I think the authors should discuss more directly how their proposed method is different from the one proposed by a partially overlapping set of authors recently (Real et al, ref 30) and why this method could not be used here.

We would like to thank the Editor and the Reviewers for their thoughtful handling of our manuscript and for the valuable and constructive comments. We believe that our manuscript has strongly profited from their suggestions. We have addressed the remaining points as detailed below.

Reviewer #2 (Remarks to the Author):

I am satisfied with the revised manuscript containing clarifications and new analyses that address my previous concerns.

Reviewer #3 (Remarks to the Author):

The authors have addressed all my comments. I think the paper will a big impact on sensory neuroscience given the rising interest in linking structure of neural circuits and function.

The only remaining suggestion I have is that I think the authors should discuss more directly how their proposed method is different from the one proposed by a partially overlapping set of authors recently (Real et al, ref 30) and why this method could not be used here.

The STNMF method presented here aims at extracting subunits without detailed constraints on the subunit shape and without the need to refer to an explicit encoding model. By contrast, the paper by Real et al. applied a full parameter optimization for a parameterized subunit model, including temporal dynamics of the subunit filters and feedback components. To handle the nonlinear and high-dimensional parameter optimization, Real et al. therefore restrained the subunits to have identical shapes and restricted the stimuli to be only one-dimensional in space, i.e., composed of stripes. The methodology of Real et al. was thus not designed for subunit identification in two spatial dimensions and with potential variability of subunit shapes. We now explain these differences between the present approach and the paper by Real et al. in the Discussion (page 11, second-to-last paragraph of Discussion).